# Noncanonical function of folate through folate receptor 1 during neural tube formation

Olga A. Balashova [1,4] ✉, Alexios A. Panoutsopoulos [1,4], Olesya Visina[1], Jacob Selhub[2], Paul S. Knoepfler [3] & Laura N. Borodinsky [1] ✉

Folate supplementation reduces the occurrence of neural tube defects (NTDs), birth defects consisting in the failure of the neural tube to form and close. The mechanisms underlying NTDs and their prevention by folate remain unclear. Here we show that folate receptor 1 (FOLR1) is necessary for the formation of neural tube-like structures in human-cell derived neural organoids. FOLR1 knockdown in neural organoids and in *Xenopus laevis* embryos leads to NTDs that are rescued by pteroate, a folate precursor that is unable to participate in metabolism. We demonstrate that FOLR1 interacts with and opposes the function of CD2-associated protein, molecule essential for apical endocytosis and turnover of C-cadherin in neural plate cells. In addition, folates increase $Ca^{2+}$ transient frequency, suggesting that folate and FOLR1 signal intracellularly to regulate neural plate folding. This study identifies a mechanism of action of folate distinct from its vitamin function during neural tube formation.

Folate is a member of the vitamin B family, participates in DNA and protein methylation and is essential for biosynthesis of nucleotides and amino acids[1]. Folate has been identified as a critical nutrient in preventing neural tube defects (NTDs)[2], prominent birth defects both in incidence and devastating consequences to fetal and infant health, by unclear mechanisms. NTDs occur when the neural tube fails to close at four weeks of pregnancy[3]. Many studies have investigated the contribution of folate in one carbon metabolism in the process of neural tube formation using diverse models, including humans[3–5]. Whether supplementation of folate periconceptionally prevents NTDs from occurring through an unknown specialized cellular and molecular mechanism in the neural tissue during neural tube formation remains unclear[6].

The utilization of folate in metabolic pathways depends on dedicated membrane proteins that facilitate its transport into cells[7]. Known folate uptake systems include: the reduced folate carrier, a classic transmembrane transporter that efficiently enables the passage of folate into the cell when folate levels are adequate; the proton-coupled folate transporter, expressed in the intestine epithelia and choroid plexus[8,9]; and the family of folate receptors (FOLRs), glycosylphosphatidylinositol-anchored proteins that implement folate uptake by receptor-mediated endocytosis[7]. Aside from folate transport, other functions of folate receptors have emerged over the last decade[6,10]. FOLR1 serves as a transcription factor in a human medulloblastoma cell line[11], interacts with proteins involved in autophagy in human hepatocellular carcinoma cell lines[12], and signals to *C. elegans* germline stem cells to proliferate[13], while FOLR2 regulates human macrophage cell adhesion to collagen[14].

FOLR1 is expressed in mouse neural folds[15] and localizes to the apical membrane of superficial neural plate cells in frog embryos[16]. FOLR1 knockout mouse[17] and knockdown frog[16] embryos exhibit NTDs. Furthermore, knockdown of FOLR1 only in the frog neural plate is enough to induce NTDs, while FOLR1 knockdown in non-neural ectoderm or mesoderm does not affect neural tube formation,

[1]Department of Physiology & Membrane Biology, Shriners Hospitals for Children Northern California, University of California Davis, School of Medicine, Sacramento, CA 95817, USA. [2]Tufts–USDA Human Nutrition Research Center on Aging, Boston, MA, USA. [3]Department of Cell Biology & Human Anatomy, Shriners Hospitals for Children Northern California, University of California Davis, School of Medicine, Sacramento, CA 95817, USA. [4]These authors contributed equally: Olga A. Balashova, Alexios A. Panoutsopoulos. ✉e-mail: oabalashova@ucdavis.edu; lnborodinsky@ucdavis.edu

indicating a tissue-specific necessity of FOLR1 expression in neural plate cells during neural tube formation[16]. The transformation of the neural plate into a tube in most vertebrates demands the folding of the former through the apical constriction of neural plate cells[3,16,18]. The understanding of this cellular event is still incomplete. It is believed that apical constriction is driven by contraction of an actomyosin network anchored to cadherins at the apicolateral adherens junctions via complexes of α- and β-catenin[19,20]. Endocytosis plays an important role during apical constriction in neural and non-neural cells by facilitating removal of plasma membrane lipids and proteins from the apical and apicolateral cell surface[21]. Several molecules have been identified as regulators of apical constriction including Shroom[18,20] and LRP2[22] in mouse and frog embryos, and FOLR1 in neurulating *Xenopus laevis* embryos[16]. The molecular mechanisms by which FOLR1 regulates neural plate cell apical constriction and whether this function is conserved across species have remained unanswered.

Here we show that folate/FOLR1 regulate neural tube formation in human-induced pluripotent stem cell (hiPSC)-based neural organoids and in *Xenopus laevis* embryos by controlling cadherin turnover in the apical adherens junctions. FOLR1 signals in neural plate cells to counteract the action of FOLR1-interacting protein CD2AP, which is necessary for endocytosis and cadherin trafficking from neural plate cell adherens junctions during apical constriction.

## Results

### A metabolically inactive folate precursor rescues FOLR1 deficiency-induced neural tube defects

To examine the role of FOLR1 in the formation of the neural tube we developed human induced pluripotent stem cell (hiPSC)-derived neural organoids[23]. This human cell-based in vitro 3D model for neural tube formation[24] is characterized by expression of the neural stem cell marker Sox2 and presence of structures with tubular morphology (Fig. 1a, b). We first assessed FOLR1 expression and discover that it localizes to apicolateral regions of neural cells surrounding the lumen of newly formed neural tube-like structures (Fig. 1a), as revealed by the proximity of FOLR1 immunostaining to α-catenin labeled apical region of neural cells surrounding the lumen. This is similar to the apical localization of FOLR1 in superficial neural plate cells of *Xenopus laevis* embryos[16], indicating similar apical targeting of FOLR1 in neural cells participating in the formation of the neural tube in the in vivo frog embryo and in the in vitro human cell-based model.

FOLR1 knockdown (FOLR1 KD, Supplementary Fig. 1a) impairs formation of neural tubes in neural organoids (Fig. 1b), resulting in structures that do not exhibit the characteristic basal displacement of nuclei surrounding the lumen (Fig. 1c), suggestive of deficient neural cell apical constriction. FOLR1-deficient cells fail to apico-basally elongate, resulting in aberrant tubular structure (Fig. 1d).

FOLR1 KD-induced phenotype is rescued by incubating 3D human neural cell-based cultures from day 0 with 50 μM pteroate (Fig. 1b–d), a folate precursor that binds to FOLR1[25] but cannot participate in metabolic pathways or be converted to folate by eukaryotic cells[13].

To demonstrate that pteroate rescues the defective neural tube-like structure phenotype through interacting with the remaining FOLR1 in the KD model, we examined the effect of knocking out (KO) FOLR1 by CRISPR/Cas9 gene editing of hiPSCs (Supplementary Fig. 1b–d). FOLR1 KO of hiPSCs results in an approximately 20% KO score (Supplementary Fig. 1b–d). Increasing the proportion of FOLR1 KO cells in neural organoids to up to 70% alters the development of neural organoids and impedes formation of neural tubes (Supplementary Fig. 2). Hence, we proceeded with the 20% FOLR1 KO (mosaic) samples in further experiments. We find that mosaic FOLR1 KO impairs formation of neural tubes (Fig. 1e), similarly to the FOLR1 KD phenotype. However, incubation with pteroate fails to rescue the FOLR1 KO-induced phenotype (Fig. 1e), in contrast to the effective rescue of the FOLR1 KD-phenotype by pteroate (Fig. 1b).

These results demonstrate that FOLR1 is necessary for the formation of human cell-derived neural tubes and suggest that folates act through FOLR1 to rescue neural tube morphogenesis by a nonmetabolic mechanism.

Similarly, pteroate partially rescues FOLR1 KD- (Supplementary Fig. 1e, f) induced NTDs in the model of neurulating *Xenopus laevis* embryos (Fig. 2a), where defects consist in failure of neural plate cell apical constriction and impairment of neural plate folding and neural tube closure[16]. Pteroate rescue of FOLR1 KD-induced NTDs mimics the rescue by folinic acid[16]. The severity of the NTD phenotype is heterogeneous, which may be related to the variability in strength of FOLR1 KD. To address this, we induced a stronger FOLR1 KD that results in more severe NTDs, characterized by degeneration of neural tissue, and find that those are not rescued by pteroate incubation (Fig. 2b). This finding mimics the lack of rescue of FOLR1-KD severe phenotype with folinic acid and is in contrast to the successful rescue of the severe phenotype from restoring FOLR1 expression in FOLR1-KD embryos[16], suggesting that a threshold level of remaining FOLR1 protein is necessary for pteroate or folinic acid rescue of the NTD phenotype.

Altogether, these discoveries demonstrate that folate/FOLR1 regulation of neural tube formation is conserved across vertebrate species and suggest a role for folate/FOLR1 independent of folate metabolism. To further identify the mechanisms by which FOLR1 knockdown impairs apical constriction and induces NTDs, we assessed histologically NTD-affected embryos and find that in addition to the open neural tube (Fig. 2c; Supplementary Fig. 3b), defects range from absence of distinguishable lumen (Supplementary Fig. 3c), lack of neural plate folding (Supplementary Fig. 3d), rounded and loosely attached neural cells (Supplementary Fig. 3d, e), as well as disruptions to neural tissue integrity (Supplementary Fig. 3e), all suggestive of deficient cell-cell adhesion. Thus, FOLR1 may enable cell-cell adhesion during neural tube morphogenesis.

### FOLR1 regulates turnover of cadherin and neural plate cell apical endocytosis

Cell-cell junctions are paramount for organizing cell movement and changes in cell shape during tissue morphogenesis. Adherens junctions, by providing cell-cell attachment and anchoring of the contractile cytoskeleton, are essential for tissue integrity during folding of the neural plate through apical constriction. We then assessed whether localization of the major adherens junction molecule cadherin is regulated by FOLR1 in hiPSC-based neural organoids. We find that FOLR1 KD or KO decreases the enrichment of cadherin in the apicolateral surface of cells surrounding the lumen, phenotype that is rescued by pteroate in KD samples (Fig. 3a) but not in KO counterparts (Fig. 3b), suggesting that a non-metabolic function of folate/FOLR1 interaction maintains cadherin-based cell-cell adhesion.

To discover the mechanisms by which FOLR1 regulates cell-cell adhesion and apical constriction, we assessed expression of C-cadherin (*cdh3*), the most abundant cadherin in the neural plate during early development of *Xenopus laevis* embryos[26], that interacts with FOLR1[16]. Results show that FOLR1 KD reduces C-cadherin protein levels (Fig. 4a). In contrast, C-cadherin transcript levels are not significantly altered by FOLR1 KD (Supplementary Fig. 4), indicating that the regulation of C-cadherin protein levels by FOLR1 is posttranscriptional.

Turnover of membrane proteins involves several steps, starting with endocytosis. In particular, removal of cadherins from adherens junctions begins with endocytosis of apicolateral membrane[27,28]. Moreover, it is believed that during apical constriction actomyosin contraction is coordinated with removal of apical cell membrane[22]. To identify the routing of C-cadherin during *Xenopus laevis* neural plate folding we assessed C-cadherin intracellular localization and find that C-cadherin-containing early endosomes (C-cadherin and EEA1 immunopositive vesicles) are immunopositive for ubiquitin (Fig. 4b, e).

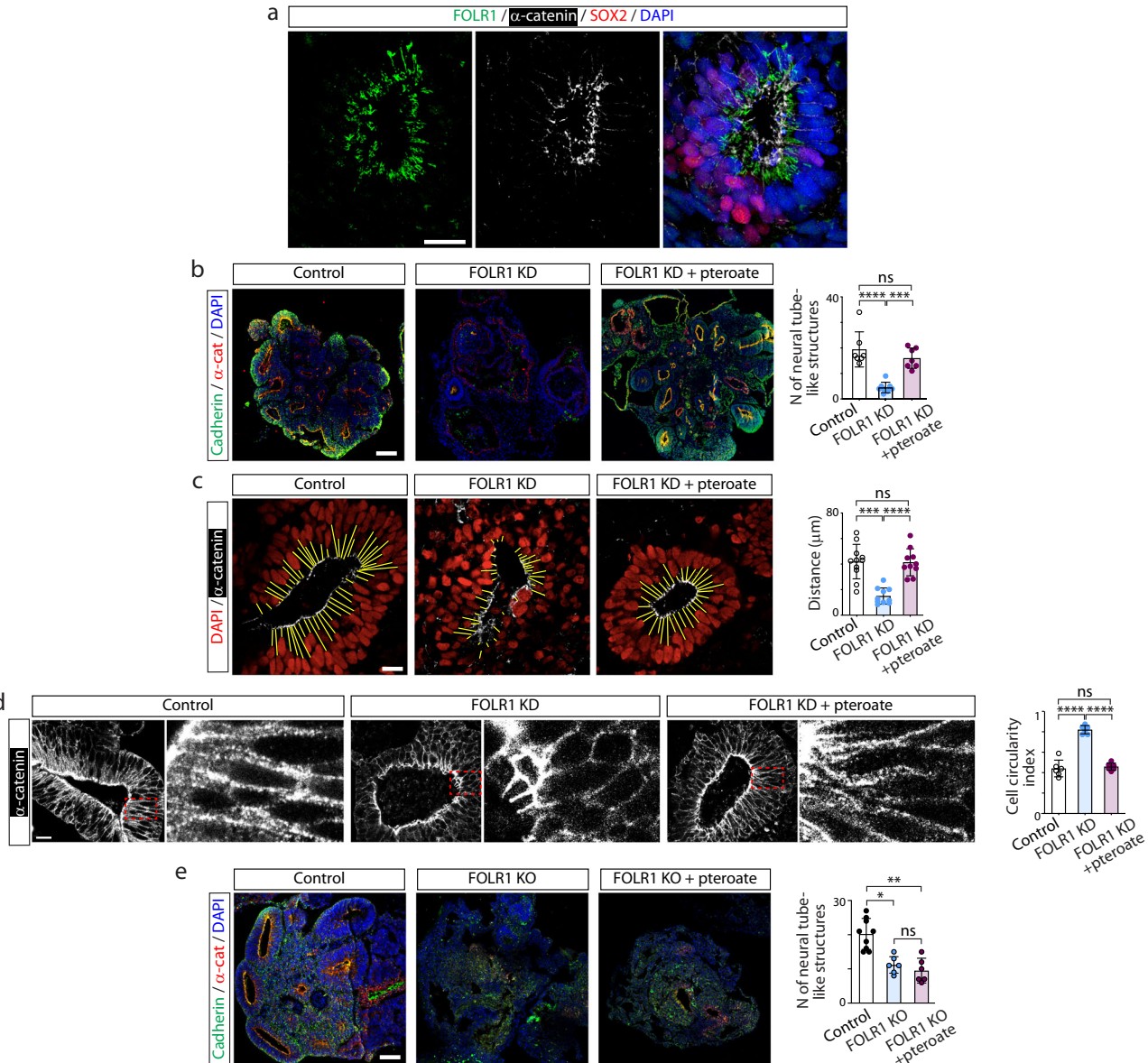

**Fig. 1 | FOLR1 regulates neural tube formation and neural cell shape in human induced pluripotent stem cell-derived neural organoids by a non-metabolic mechanism.** Neural organoids were generated from human induced pluripotent stem cells (hiPSCs). After 18–21-days in vitro neural organoids were fixed, sliced and processed for immunostaining and nuclear labeling. **a** FOLR1 localizes to the apical surface of neural cells surrounding the neural tube-like structure lumen. Shown is a maximum intensity projection. Similar pattern of localization was observed in $n = 15$ neural organoids from $N = 3$ independent experiments. **b–d** hiPSC-derived embryoid bodies were incubated with 2 µM control-*vivo*-morpholino (Control), FOLR1-vivo morpholino (FOLR1 knockdown (KD), FOLR1-MO) or FOLR1-MO and 50 µM pteroate (FOLR1 KD + pteroate) until 18–21 days in vitro. **b** Examples of immunostained neural organoids in control and experimental groups. Graph shows individual and mean ± SD number of neural tube-like structures per 10 µm-thick organoid section, $n = 7$, 8 and 7 neural organoids for Control, FOLR1 KD and FOLR1 KD+pteroate groups, respectively from $N = 3$ independent experiments. **c, d** Examples of immunostained neural tube-like structures. **c** Yellow lines indicate distance from lumen border to first layer of nuclei. Graph shows individual and mean ± SD lumen-nuclei distance per neural tube-like structure, $n = 12$, 11 and

11 neural organoids for Control, FOLR1 KD and FOLR1 KD+pteroate groups, respectively from $N = 3$ independent experiments. **d** α-catenin immunostaining was used to define cellular contour and measure circularity index. Dashed outlined boxes correspond to zoomed-in images. Graph shows individual and mean ± SD circularity index (1: perfect circle; 0: elongated polygon) per neural tube-like structure, $n = 35$ cells per neural tube, $n = 6$ neural tubes per group in $N = 3$ independent experiments. **e** Cultured hiPSCs reaching 75% confluency were transfected with FOLR1-sgRNA/CRISPR/Cas9 (FOLR1 KO) and used for neural organoid cultures. Cultures were supplemented with vehicle or 50 µM pteroate (FOLR1 KO + pteroate) until 18–21 days in vitro. Shown are examples of immunostained neural organoids in control and experimental groups. Graph shows individual and mean ± SD number of neural tube-like structures per 10 µm-thick organoid section, $n = 9$, 6 and 6 neural organoids for Control, FOLR1 KO and FOLR1 KO+pteroate groups, respectively from $N = 3$ independent experiments. In (**b–e**), $*p < 0.05$, $**p < 0.01$, $***p < 0.001$, $****p < 0.0001$, ns not significant, one-way ANOVA, Tukey post-hoc multiple comparison test. Scale bars are 20 (**a, c–d**) or 100 (**b, e**) µm. Source data are provided as a Source Data file.

Intracellular C-cadherin also partially co-localizes with late endosomal and lysosomal markers Rab7 and LAMP1, respectively, (Fig. 4c–e), indicating that endocytosed C-cadherin is targeted for degradation. We then examined in vivo endocytosis in the apical surface of FOLR1-

depleted neural plate cells during early stages of neural plate folding by expressing the fluorescently tagged early endosomal marker EEA1, that colocalizes with endogenous C-cadherin (Supplementary Fig. 5), followed by live-imaging of the apical surface of neural plate cells in

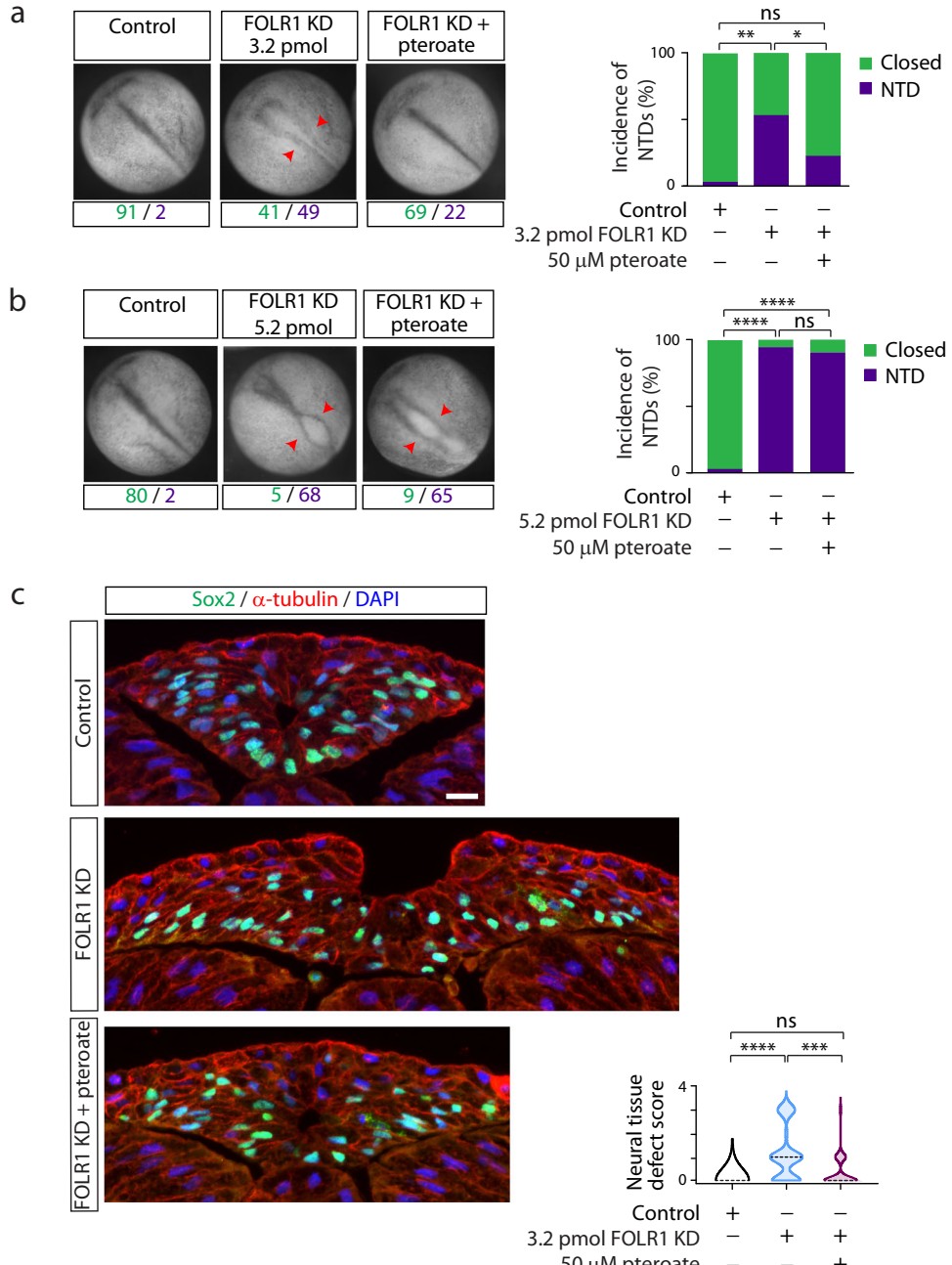

**Fig. 2 | FOLR1 is necessary for neural tube formation in *Xenopus laevis* embryos through a non-metabolic mechanism.** Two-cell stage *Xenopus laevis* embryos were microinjected with 3.2 (**a**, **c**) or 5.2 (**b**) pmol Control-MO (Control) or FOLR1-MO (FOLR1 KD) per embryo and incubated with saline or 50 μM pteroate until the neural tube closed in control embryos, when they were fixed, photomicrographed (**a**, **b**), sectioned and processed for immunostaining (**c**). **a**, **b** Examples of embryos in each group. Arrowheads indicate open neural tube (neural tube defect, NTD). Numbers are embryos presenting open (NTD, purple) or closed (green) neural tube. Bar graphs represent proportion of phenotypes in each group. **c** Examples of immunostained transverse sections of the neural tissue; n of embryos sectioned was 25, 39 and 42 in Control, FOLR1 KD, and FOLR1 KD+pteroate groups, respectively, in *N* = 3 independent experiments. Scale bar is 20 μm. Graph shows distribution of neural tissue defect score per group, median is indicated by dashed line. In (**a**–**c**), *$p < 0.05$, **$p < 0.01$, ***$p < 0.001$, ****$p < 0.0001$, one-way ANOVA, Tukey post-hoc multiple comparison test (**a**, **b**), or Kruskal-Wallis, Dunn's multiple comparison test (**c**). Source data are provided as a Source Data file.

embryos with unilaterally downregulated FOLR1. Results show that embryos exhibit increased endocytosis in the FOLR1 KD neural plate apical surface at the onset of apical constriction (Fig. 4f). These findings suggest that surplus apical endocytosis in FOLR1-deficient neural tissue may cause excessive removal of C-cadherin from apicolateral adherens junctions and promote its degradation.

Vascular endothelial cadherin undergoes proteolytic cleavage of the cytosolic domain during endocytosis, thereby biasing cadherin towards postendocytic degradation over recycling[29]. We find that inhibition of protein degradation in neurulating *Xenopus* embryos results in accumulation of C-cadherin N-terminal fragment in the neural plate (Supplementary Fig. 6), suggesting that proteolytic processing of C-cadherin precedes its degradation. We discover that levels of C-cadherin N-terminal fragment are significantly increased in FOLR1 KD neural plate (Fig. 5a) along with increased C-cadherin ubiquitination (Fig. 5b).

These data suggest that FOLR1 negatively regulates endocytosis and C-cadherin degradative trafficking to preserve adequate levels of

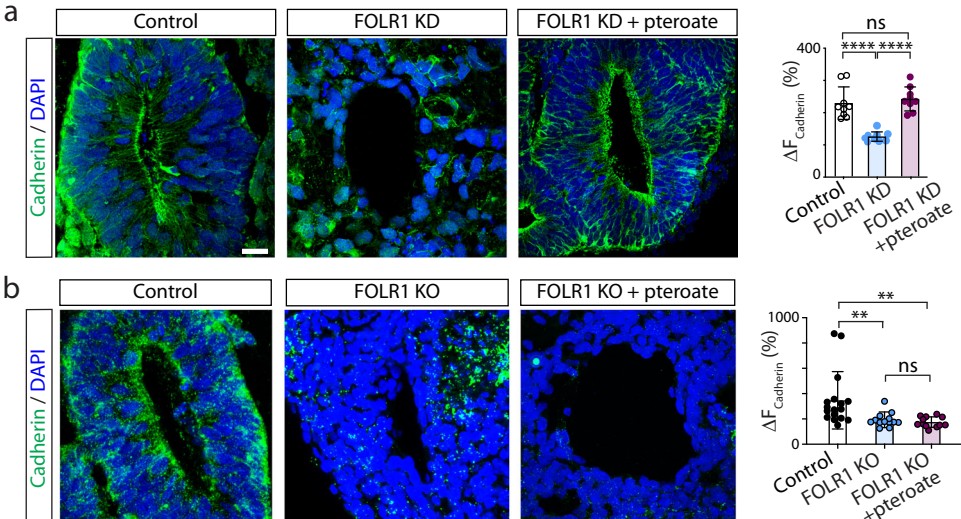

**Fig. 3 | FOLR1 enables cadherin enrichment in apicolateral neural cells surrounding the lumen of neural tube-like structures in hiPSC-derived neural organoids by a non-metabolic mechanism. a** hiPSC-derived embryoid bodies were incubated with 2 μM control-*vivo*-morpholino (Control), FOLR1-*vivo* morpholino (FOLR1 knockdown (KD), FOLR1-MO) or FOLR1-MO and 50 μM pteroate (FOLR1 KD + pteroate) until 18–21 days in vitro when neural organoids were fixed and processed for immunostaining and nuclear labeling with DAPI. **b** Cultured hiPSCs reaching 75% confluency were transfected with FOLR1-sgRNA/CRISPR/Cas9 (FOLR1 KO) and used for neural organoid cultures. Cultures were supplemented with vehicle (Control) or 50 μM pteroate (FOLR1 KO + pteroate) until 18–21 days in vitro when neural organoids were fixed and processed for immunostaining and nuclear labeling with DAPI. **a**, **b** Images are examples of immunostained neural tube-like structures. Graphs show individual and mean ± SD maximum percent change in cadherin immunolabeling fluorescence intensity when crossing the lumen per neural tube-like structure, $n = 9$ (**a**) and $n = 12$ (**b**) neural organoids per group, $N = 3$ independent experiments. **$p < 0.01$, ***$p < 0.0001$, ns not significant, one-way ANOVA, Tukey post-hoc multiple comparison test. Scale bars are 20 μm. Source data are provided as a Source Data file.

cell adherens junction complexes for efficient neural plate cell apical constriction.

## FOLR1 interacts with CD2AP and counteracts its action in the neural plate

FOLR1 is a GPI-anchored protein attached to the extracellular leaflet of the cell membrane. Thus, regulation of C-cadherin intracellular turnover by FOLR1 may involve molecular partners. To identify potentially novel candidate proteins interacting with FOLR1 we performed a proteomic analysis of FOLR1-immunoprecipitates from neural plate-stage *Xenopus laevis* embryos through LC-MS/MS. Results confirm the previously identified interaction between FOLR1 and cell adherens junction molecules C-cadherin and β-catenin[16] and revealed several other candidate proteins, among them CD2-associated protein (CD2AP, Supplementary Table 1). We confirmed FOLR1 interaction with CD2AP through Western blot assays of FOLR1-immunoprecipitates (Fig. 6a).

CD2AP is an adaptor protein that connects membrane proteins to the actin cytoskeleton[30,31] and promotes ubiquitination and endocytosis of receptor tyrosine kinases[32–35]. It has a paralogous gene *sh3kbp1* coding for SH3-domain kinase binding protein 1, also known as CIN85 (CBL-interacting protein of 85 kDa), as a result of a duplication event early in vertebrate evolution[36]. Unlike mice, which express both during neural tube formation[31], *Xenopus laevis* embryos only express *cd2ap*[37]. Like its paralog[34,35], CD2AP interacts with E3 ubiquitin-protein ligase CBL, which targets membrane proteins for lysosomal degradation[38–40]. Interestingly, in non-neural epithelial cells CBL-like ubiquitin ligase Hakai ubiquitinates epithelial cadherin in the adherens junction complex, promoting E-cadherin endocytosis[41]. Immunostaining assays demonstrate that similarly to FOLR1 (Fig. 1a and[16]), p-CD2AP and p-CBL localize at the apical membrane of superficial neural plate cells during neural plate folding (Fig. 6b). CD2AP knockdown (KD, Supplementary Fig. 7) results in NTDs (Fig. 6c) associated with failure of neural plate cells to apically constrict (Fig. 6d), similar to the phenotypes resulting from FOLR1 KD (Fig. 1b−g and[16]). Next, we assessed endocytosis in the CD2AP KD apical neural plate and, intriguingly, find that in contrast to FOLR1 KD

(Fig. 4f), CD2AP downregulation reduces endocytosis (Fig. 7a). Moreover, CD2AP KD increases C-cadherin protein level (Fig. 7b). These results argue that reduced endocytosis may interfere with the degradation of C-cadherin in CD2AP-deficient neural plates. Our findings suggest that CD2AP promotes endocytosis and C-cadherin removal from adherens junctions during apical constriction.

Thus, FOLR1 and CD2AP emerge as important and opposing regulators of neural plate cell apical endocytosis and cell adherens junction turnover. Hence, we investigated whether FOLR1 may affect these processes by controlling CD2AP protein level. We find that FOLR1 downregulation increases CD2AP protein level in the neural plate (Fig. 8a). Interestingly, CD2AP upregulation in FOLR1 KD neural plates did not occur when inhibitors of protein degradation were used (Fig. 8a), indicating that FOLR1 may target CD2AP for degradation.

Reciprocally, CD2AP KD upregulates FOLR1 protein levels, effect that is abolished in the presence of proteasome and lysosome inhibitors during neural plate folding (Fig. 8b). This result can be explained by the inhibitory effect of CD2AP KD on apical membrane endocytosis, where FOLR1 localizes. Hence, diminished FOLR1 endocytosis when CD2AP is depleted may halt FOLR1 turnover upregulating FOLR1 protein levels.

Altogether these discoveries indicate that FOLR1 regulates CD2AP protein level to achieve a balanced remodeling of the apicolateral cell membrane by regulating the rate of apical endocytosis and adherens junction turnover necessary for adequate and precisely timed neural plate cell apical constriction during neural tube morphogenesis.

## Folates increase Ca²⁺ activity in the neural plate

Apical constriction operates through pulses of spontaneous $Ca^{2+}$ transients and actomyosin contraction to enable neural tube formation[42]. $Ca^{2+}$ transient frequency increases with the progression of neural plate folding in *Xenopus laevis* embryos[43], suggesting a progressive engagement of $Ca^{2+}$ signaling during neural tube formation. Indeed, interfering with $Ca^{2+}$ dynamics during neural tube formation results in NTDs in diverse species, including, chick, mouse

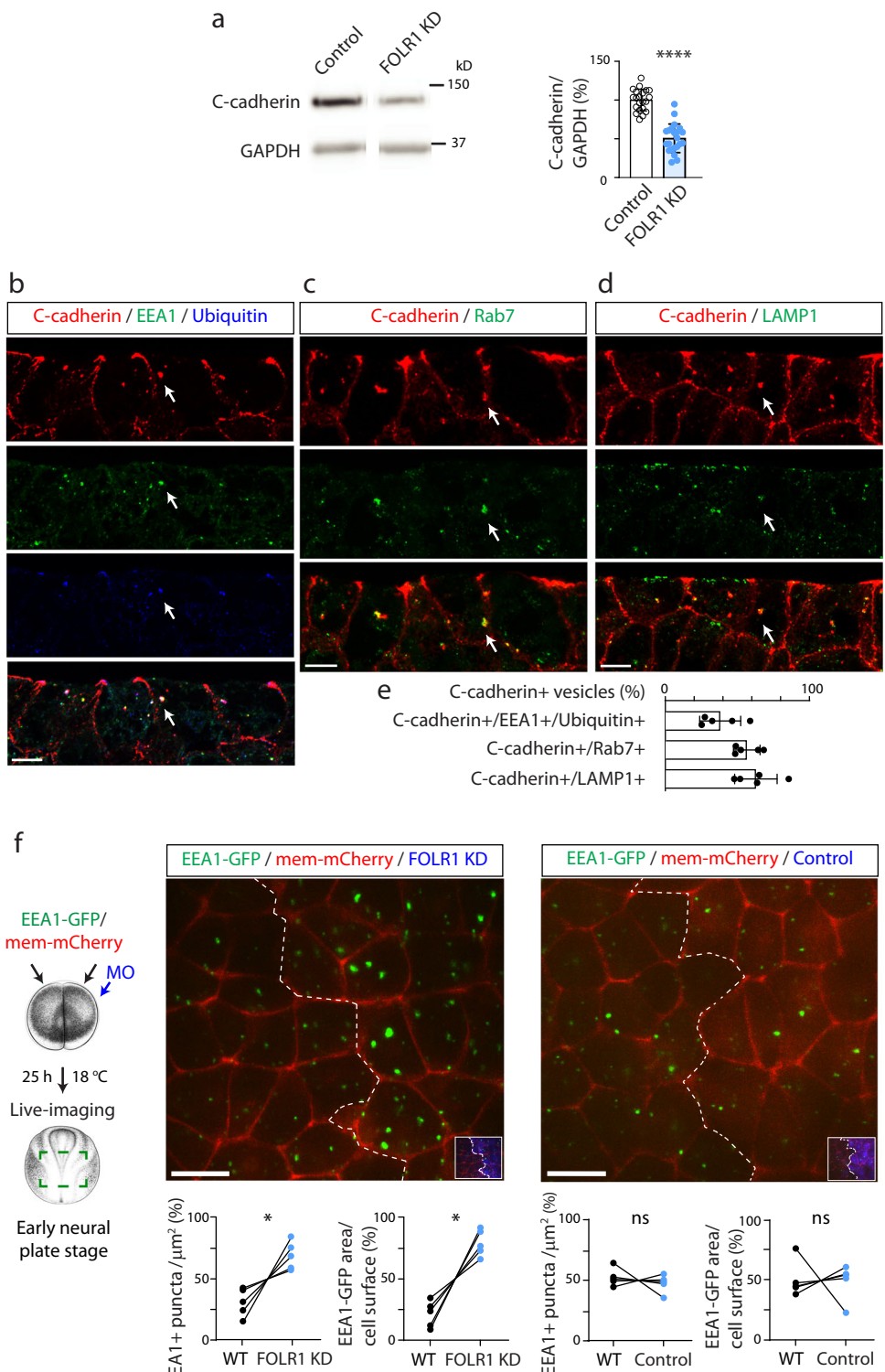

and amphibians[43–46]. Moreover, several Ca²⁺-regulated signaling molecules, including the proteolytic enzyme calpain, calmodulin and PKC, regulate cytoskeletal dynamics of neural plate cells and neural tube formation in developing embryos[45,47,48]. To further uncover the signaling mechanisms that folate/FOLR1 elicit in the embryonic neural tissue, acutely dissociated neural cells from mid neural plate stage embryos were plated and loaded with a cell-permeant Ca²⁺-sensitive dye, Fluo4-AM, followed by timelapse imaging before and after addition of different concentrations of folic or folinic acid. Results show that folates elicit acute Ca²⁺ transients in a

concentration-dependent manner with kinetics comparable to the spontaneously occurring transients in cultured neural plate cells (Fig. 9a–c).

To determine whether this folate-triggered signaling is present in vivo during neural plate folding, a genetically encoded Ca²⁺ sensor, GCaMP6s, was expressed in developing embryos. We find that incubation with folinic acid increases the frequency of Ca²⁺ transients during neural plate folding and not at earlier, prior to apical constriction, stages (Fig. 9d). In contrast, down-regulating FOLR1 expression decreases the frequency of Ca²⁺

**Fig. 4 | FOLR1 regulates C-cadherin protein level and endocytosis from apical neural plate cell membrane. a** Two-cell stage *Xenopus laevis* embryos were bilaterally microinjected with 1.6 pmol Control-MO (Control) or FOLR1-MO (FOLR1 KD) per blastomere and allowed to grow until they reached mid-neural plate stages (stage 15–17) when neural plate was dissected and processed for Western blot assays. Example of Western blot assay. Graph shows individual and mean ± SD percent of optical density (OD) for C-cadherin immunoblot band normalized with GAPDH protein band OD and compared to controls. Two-tailed ratio *t*-test, *n* = 42 and 44 neural plates for Control and FOLR1 KD groups, respectively, *N* = 8 independent experiments. **b–e** Neural plate stage *Xenopus laevis* embryos were fixed, sectioned and processed for immunostaining. **b–d** Examples of immunostained neural plate. Arrows indicate colocalization of C-cadherin with early endosomal marker (EEA1) and ubiquitin (**b**), late endosomal marker (Rab7, **c**) and lysosomal marker (LAMP1, **d**). Scale bar, 10 µm. **e** Graph shows mean ± SD proportion of C-cadherin vesicles colocalizing with the indicated markers. Number of cells analyzed were 104, 52, 65 from immunostained samples in (**b**, **c** and **d**), respectively,

from *N* = 5 embryos. **f** Two-cell stage *Xenopus laevis* embryos were bilaterally microinjected with hEEA1-GFP mRNA and membrane mCherry and unilaterally microinjected with 2.6–3.2 pmol FOLR1-MO (FOLR1 KD) or Control-MO (Control) per blastomere along with fluorescent tracer and allowed to develop until they reached early neural plate stages (stage 13-14), when they were time-lapse imaged with an acquisition rate of 1 frame/6 min. Schematic shows experimental design. *Xenopus* illustrations © Natalya Zahn (2022) obtained from Xenbase (www.xenbase. org RRID:SCR_003280), released under a Creative Commons Attribution-NonCommercial 4.0 License (CC BY-NC)[68]. Images are maximum intensity projection of single time frame. Dashed lines indicate border between wild-type (WT) and MO-injected neural plate. Insets show neural plate injected side with tracer in blue. Graphs show distribution between both halves of the neural plate (in %) of the number of EEA1-GFP vesicles and area fraction of labeled endosomes per neural plate cell surface. Two-tailed paired *t*-test; ns: not significant, *n* = 30 cells analyzed in each group, *N* = 5 embryos per group. Scale bar, 20 µm. In **a**, **f**, *p < 0.05, ****p < 0.0001. Source data are provided as a Source Data file.

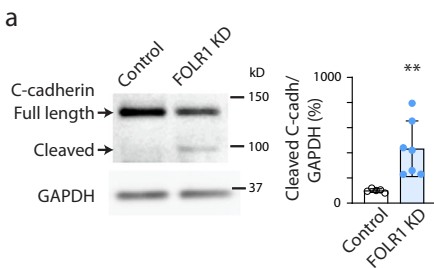
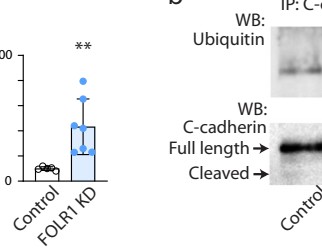
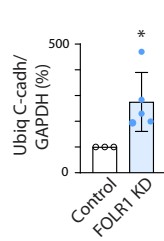

**Fig. 5 | FOLR1 downregulation enhances C-cadherin cleavage and ubiquitination in *Xenopus laevis* neural plate.** Two-cell stage *Xenopus laevis* embryos were bilaterally microinjected with 1.6 pmol Control-MO (Control) or FOLR1-MO (FOLR1 KD) per blastomere, incubated from early neural plate stage (stage 12) with vehicle or proteasome and lysosome inhibitors until they reached mid-neural plate stages (stage 16–17) when neural plates were dissected and processed for Western blot (**a**, **b**) and immunoprecipitation (**b**) assays. **a** Example of Western blot assay immunoprobed for C-cadherin showing full-length and cleaved (~80 kD) forms. Graph shows individual and mean ± SD percent of optical density (OD) for cleaved C-cadherin band normalized with GAPDH protein band

OD and compared to controls, *n* = 20 and 26 neural plates for Control and FOLR1 KD groups respectively, *N* = 3 independent experiments. **b** Example of immunoprecipitation (IP) assay for C-cadherin followed by Western blot assay for ubiquitin and C-cadherin. Graph shows individual and mean ± SD percent of optical density (OD) for ubiquitinated (Ubiq) C-cadherin band normalized with full-length C-cadherin and compared to controls. In (**a**, **b**), *p < 0.05, **p < 0.01, two-tailed ratio *t*-test, *n* = 16 and 24 neural plates for Control and FOLR1 KD groups respectively, *N* = 3 independent experiments. Source data are provided as a Source Data file.

transients (Fig. 9e), suggesting that FOLR1 elicits Ca²⁺ dynamics during neural plate folding.

Ca²⁺ dynamics in the neural plate are dependent on multiple mechanisms, including NMDA receptors[43] and L-type voltage-gated Ca²⁺ channels[46]. We examined the mechanisms of folate-mediated Ca²⁺ transients in the folding neural plate and found that a cocktail of Na⁺ and voltage-gated Ca²⁺ channel blockers inhibits both spontaneous and folic acid-induced Ca²⁺ transients (Fig. 9f–h), suggesting that folate/FOLR1-dependent Ca²⁺ dynamics operate through Ca²⁺ influx.

Altogether these finding suggest that folate/FOLR1 recruit rapid signaling mechanisms important for regulating neural plate cell apical constriction and neural tube formation.

## Discussion

Endocytosis of membrane and removal of cell-cell adhesion proteins from the cell apical circumference are essential for the reduction of apical surface. At the same time, adequate level of adherens junction complexes is required throughout the course of apical constriction to anchor actomyosin cytoskeleton and generate sufficient contraction force. The molecular mechanisms that ensure balanced apical trafficking during neural cell apical constriction have remained unclear. This study shows that endocytosis and C-cadherin protein levels in the neural plate are regulated by FOLR1 and CD2AP such that FOLR1 impedes endocytosis, C-cadherin cleavage and ubiquitinition, favoring retention of C-cadherin at the apicolateral adherens junctions. In contrast, CD2AP promotes endocytosis and C-cadherin protein turnover,

enabling trafficking from the apical neural cell circumference. We propose a model in which FOLR1 by targeting CD2AP for degradation, controls the rate of apical trafficking, to achieve adequate levels of adherens junction proteins for actomyosin contractility. The counteracting actions of FOLR1 and CD2AP on cadherin turnover, in turn, orchestrates appropriate spatiotemporal changes in cell-cell adhesion to enable apical constriction during neural plate folding (Fig. 9i).

CD2AP is a known regulator of cell-cell adhesion and endocytosis; first discovered in the context of T-cell and antigen-presenting cell interaction[30]. Further studies showed that in the kidney CD2AP associates with nephrin, the primary component of the slit diaphragm, specialized junction of glomerular epithelial cells[31]. Regulation of CD2AP levels by proteolytic program is responsible for TGFβ-dependent kidney podocyte ultrafiltration function, while a dysregulation of CD2AP levels constitutes the molecular mechanism underlying proteinuric kidney disease[49]. Moreover, mice lacking CD2AP exhibit a congenital nephrotic syndrome, indicating a role for CD2AP in kidneys development[31]. Gastrointestinal tract development is also dependent on CD2AP expression, because knockout mice exhibit mislocalization of E-cadherin resulting in disturbed adherens junctions altering the morphogenesis of the gastric unit[50]. CD2AP action is regulated by its phosphorylation[51], association with other molecules in complexes[38,40], and by protein turnover[40]. Here we show that FOLR1 negatively regulates CD2AP function by controlling CD2AP protein degradation during neural plate folding. Identifying the mechanisms by which FOLR1 regulates CD2AP protein turnover demands further

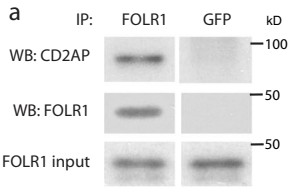

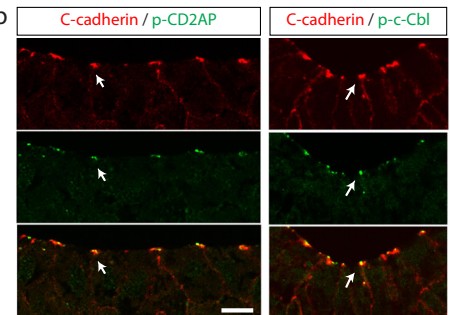

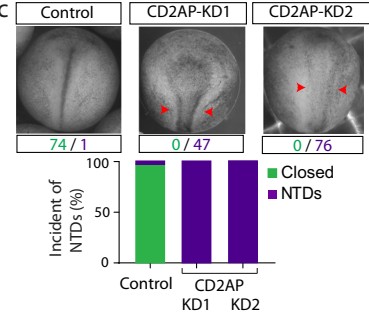

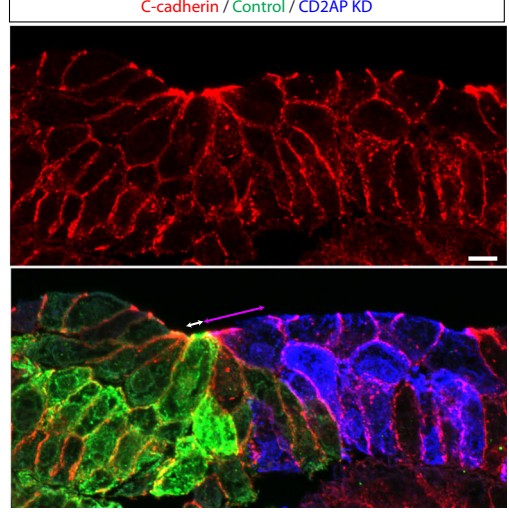

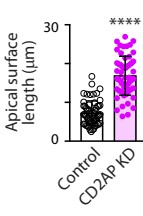

**Fig. 6 | CD2AP interacts with FOLR1 and is necessary for apical constriction and neural tube formation. a** Neural plate stage *Xenopus laevis* embryos were processed for co-immunoprecipitation (IP) assays. Example of Western blot assay from immunoprecipitates with FOLR1 or GFP (control) antibodies and probed for CD2AP or FOLR1. Similar results were observed in $N = 3$ independent experiments. **b** Neural plate stage *Xenopus laevis* embryos were fixed and processed for immunostaining. Images are transverse single z-sections of immunostained neural plate showing apical colocalization of phospho-CD2AP (p-CD2AP) and p-c-Cbl with C-cadherin. Scale bar, 10 μm. Similar results were observed in $N = 3$ independent experiments. **c** Two-cell stage *Xenopus laevis* embryos were bilaterally microinjected with 9.9 pmol of Control-morpholino (Control, Control-MO), 2.6 pmol CD2AP-MO1 (CD2AP KD1) or 7.4–9.9 pmol CD2AP-MO2 (CD2AP KD2/KD) per blastomere until neural tube closed in control embryos, when they were fixed and photo-micrographed. Examples of whole embryos in each group. Arrowheads indicate

open neural tube (neural tube defect, NTD). Numbers are embryos presenting closed (green) or open (purple, NTD) neural tube. Bar graph represents proportion of phenotypes in each group. **d** Two-cell stage *Xenopus laevis* embryos were uni-laterally microinjected with 9.9 pmol Control-MO (Control) and 7.4–9.9 pmol CD2AP-MO2 (CD2AP KD) along with GFP and mCherry mRNA, respectively, and allowed to develop until they reached mid-neural plate stages, when they were fixed and processed for immunostaining. Image is a transverse section of the neural plate, immunostained for GFP (Control), mCherry (CD2AP KD) and C-cadherin. Double arrows indicate apical surface length of medial superficial Control (white) and CD2AP KD (magenta) neural plate cells. Scale bar, 20 μm. Graph shows individual and mean ± SD apical length of superficial neural plate cells per embryo, *n* of cells = 75 in each half of the neural plate, *N* of embryos = 4. ****$p < 0.0001$, two-tailed paired *t*-test. Source data are provided as a Source Data file.

investigation. Folate-FOLR1 interaction may recruit a transmembrane signaling partner that in turn regulates CD2AP-Cbl-mediated endocytosis, ubiquitination and turnover of adherens junction molecules. Indeed, we find that folates trigger acute changes in $Ca^{2+}$ dynamics in the neural plate, which are also dependent on FOLR1 expression. Folate/FOLR1-dependent $Ca^{2+}$ dynamics may serve as the link between the regulated removal of apicolateral membrane and the cytoskeletal dynamics necessary for neural plate cell apical constriction during neural plate folding.

The dependence of the process of neural plate folding on increases in intracellular $Ca^{2+}$ concentration has long been recognized.

Neurulating amphibian embryos in which spontaneous intracellular $Ca^{2+}$ increases were blunted, resulting in halted formation of neural ridges and neural folds, are rescued by incubation with an ionophore[44]. Further studies demonstrated that in chick embryos $Ca^{2+}$ influx-triggered neural plate folding is mediated by $Ca^{2+}$-induced neural plate cell apical constriction in a calmodulin-dependent manner[45]. In mouse embryos knockout of MacMARCKS, a PKC substrate that binds F-actin in a $Ca^{2+}$/calmodulin and PKC-sensitive manner, leads to cranial neural tube defects[47]. More recent studies demonstrate that, increases in intracellular $Ca^{2+}$ concentration are transient, spontaneous, asynchronous and mostly occurring in single cells in *Ciona savignyi*[52] and

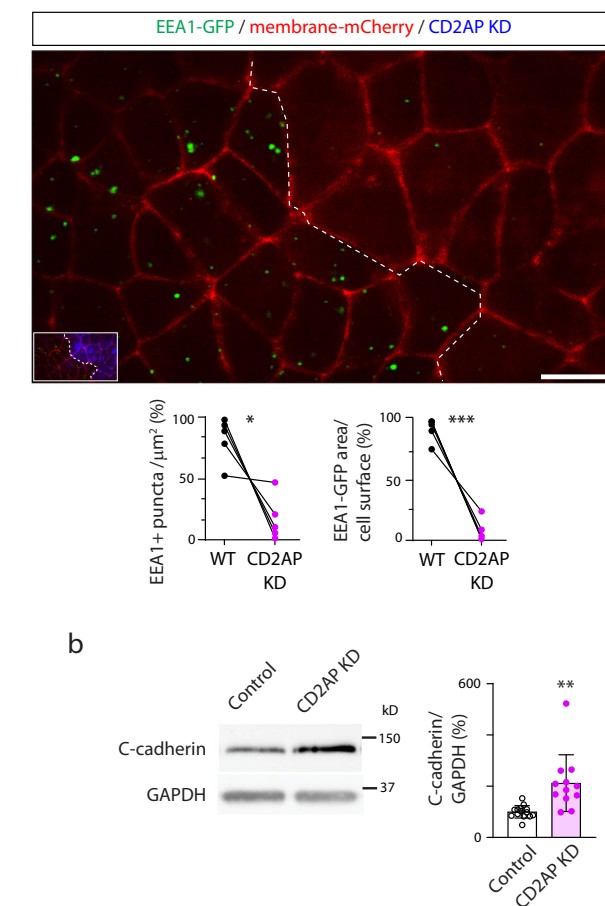

**Fig. 7 | CD2AP is necessary for apical endocytosis and C-cadherin turnover in the neural plate. a** Two-cell stage *Xenopus laevis* embryos were bilaterally micro-injected with hEEA1-GFP and membrane mCherry mRNAs and unilaterally micro-injected with 9.9 pmol CD2AP-MO (CD2AP KD) per blastomere along with fluorescent tracer and allowed to develop until they reached early neural plate stages (stage 13-14), when they were time-lapse imaged with an acquisition rate of 1 frame/6 min. Image is maximum intensity projection of single time frame. Dashed line indicates border between morpholino-injected and wild-type (WT) neural plate. Inset shows neural plate injected side showing tracer in blue. Graphs show distribution between both halves of the neural plate (in %) of the number of EEA1-GFP vesicles and area fraction of labeled endosomes per neural plate cell surface. Two-tailed paired *t*-test, *n* = 28 cells analyzed in each group from *N* = 5 embryos per group. Scale bar, 20 μm. **b** Two-cell stage *Xenopus laevis* embryos were bilaterally microinjected with 7.4 pmol Control-MO (Control) or CD2AP-MO (CD2AP KD) per blastomere and allowed to grow until they reached mid-neural plate stages (stage 15–17) when neural plate was dissected and processed for Western blot assays. Image is an example of Western blot assay. Graph shows individual and mean ± SD percent of optical density (OD) for C-cadherin immunoblot band normalized with GAPDH protein band OD and compared to controls. Two-tailed ratio *t*-test, *n* = 28 and 24 neural plates for Control and CD2AP KD groups, respectively, *N* = 5 independent experiments. In (**a**, **b**), *\*p* < 0.05, \*\**p* < 0.01, \*\*\**p* < 0.001. Source data are provided as a Source Data file.

*Xenopus laevis*[42,52] neural plate. These Ca$^{2+}$ transients are dependent on Ca$^{2+}$ influx[46,52] and Ca$^{2+}$ release from intracellular stores[46]. The dependence on appropriate Ca$^{2+}$ dynamics for actomyosin contractility during apical constriction in the neural plate was also demonstrated in the mouse embryo in which loss of function of *Spca1*, ATPase that regulates Ca$^{2+}$ homeostasis, results in NTDs due to impaired myosin II and cofilin 1 localization[53]. As for the cell membrane partners and downstream targets of folate/FOLR1-dependent Ca$^{2+}$ transients, further investigation is needed. Regulation of Ca$^{2+}$-sensitive proteases, such as members of the calpain family, could be potential downstream

effectors, given that they are Ca$^{2+}$-sensitive and necessary for neural tube formation[42]. Alternatively, folate/FOLR1-modulated Ca$^{2+}$ dynamics may recruit signaling molecules that target CD2AP for degradation. Supporting this is the Ca$^{2+}$/calmodulin-dependent association of the ubiquitin ligase Cbl with protein complexes regulating RTK function in tumor cell lines[54] and, reciprocally the RTK signaling-dependent regulation of CD2AP levels[40]. The precise relationship and timing of folate/FOLR1-dependent Ca$^{2+}$ signaling and adherens junction remodeling during apical constriction needs to be further investigated. Despite a single Ca$^{2+}$ transient may be a faster event than the regulation of the endocytic process during apical constriction, the information carried by Ca$^{2+}$ dynamics in neural cells may be integrated over time and be transduced into regulating apical endocytosis and degradation of adherens junction components.

This study demonstrates a non-canonical mechanism of folate/FOLR1 action in the frog embryo and in an in vitro human cell-based model that shifts the long-standing paradigm of FOLR1 acting exclusively as a folate transporter for folate participation in one-carbon metabolism. Our findings raise the possibility that folate via FOLR1 regulates apical membrane and cadherin removal necessary for apical constriction and neural tube formation. Pteroate rescue of only milder KD-induced NTDs in frog embryos and that of FOLR1 KD but not KO in hiPSC-derived neural organoids supports the idea that it enhances the function of the remaining FOLR1 from the knock-down through a non-metabolic mechanism (Fig. 9i). This non-metabolic function resembles the role folates have in the uni-cellular organism *Dictyostelium*, where folate has been long recognized as a signaling molecule that acts as a chemoattractant and regulator of phagocytosis and macropinocytosis through its binding to the amoeba's folate receptor that elicits Ca$^{2+}$ transients to change cytoskeletal dynamics[55–57]. Nevertheless, the amoeba's folate receptor is a G-protein coupled receptor, unlike in vertebrates where FOLR1 is a GPI-anchored protein. This poses the interesting consideration on whether folate receptor evolved from the self-contained signaling receptor to a form that needs to partner with other transmembrane and intracellular molecules to elicit signaling. A non-metabolic function for folate through FOLR1 has also been shown in C. elegans germline cells, where pteroate regulates cell proliferation[13].

A full understanding of the mechanisms of action of folate and FOLR1 during neural tube formation may contribute to developing effective preventative strategies for NTDs and to identifying potential risk factors.

## Methods

All experimental procedures and research design utilized in this study complied with ethical regulations. The Institutional Animal Care and Use Committee approved the animal protocol #22264 implemented in this study. IACUC follows the guidelines established by the Animal Welfare Act and the Public Health Service Policy on Humane Care and Use of Laboratory Animals.

### Generation of neural organoids from human induced pluripotent stem cells

Human wild type induced pluripotent stem cells (hiPSCs; WTC-11 hiPSC line, Allen Institute, GM25256, Coriell Institute for Medical Research) were used to generate neural organoids according to protocols developed by Lancaster et al.,[23]. Briefly, after hiPSC colonies reached 70–80% confluency in feeder independent conditions in mTESR medium (Stem Cell Technology, cat. # 85850), they were detached with Accutase (Stem Cell Technology, cat. # 07920). The mixture was triturated to obtain single cells and centrifuged for 3 min at 270 *g* at 23 °C. Cells were resuspended in hESC medium, prepared as following for 500 ml: 100 ml KnockOut Serum Replacement (Thermofisher, cat. # 10828010), 15 ml ESC-quality FBS

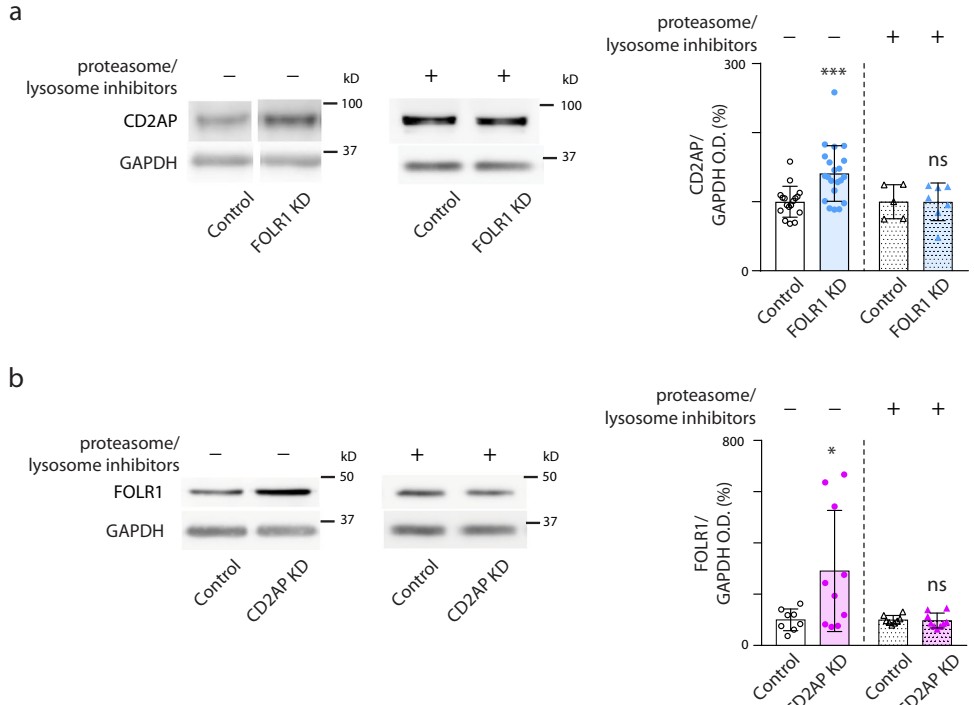

**Fig. 8 | FOLR1 and CD2AP reciprocally regulate their protein turnover.** Two-cell stage *Xenopus laevis* embryos were microinjected with 14.8 pmol Control-morpholino (MO, Control, **a**, **b**), 3.2 pmol FOLR1-MO (FOLR1 KD, **a**) or 14.8 pmol CD2AP-MO2 (CD2AP KD, **b**) per embryo and incubated with saline or proteasome and lysosome inhibitors at the end of gastrulation (stage 12) until neural plate stages (stage 17) when they were processed for Western blot assays. Images are examples of Western blot assays. Graphs show individual and mean ± SD percent of optical density (OD) for CD2AP (**a**) or FOLR1 (**b**) immunoblot band normalized with GAPDH protein band OD and compared to controls. In (**a**), *n* = 34 and 42 neural plates for Control and FOLR1 KD, respectively, *N* = 7 independent experiments; *n* = 20 and 26 neural plates for Control+inhibitors and FOLR1 KD+inhibitors groups, respectively, *N* = 3 independent experiments. In (**b**) *n* = 16 and 20 for Control and CD2AP KD groups, respectively and *n* = 16 and 18 neural plates for Control+inhibitors and CD2AP KD+inhibitors, respectively, *N* = 3 independent experiments. In (**a**, **b**) *\*p* < 0.05, *\*\*\*p* < 0.001, ns: not significant, two-tailed ratio *t*-test. Source data are provided as a Source Data file.

(Thermofisher, cat. # 1047901), 5 ml GlutamMAX (Thermofisher, cat. # 35050061), 5 ml MEM-NEAA (Millipore-Sigma, cat. # M7145), 3.5 µl β-mercaptoethanol, up to 500 ml with DMEM-F12 (Thermofisher, cat. # 11330032) with 50 µM ROCK inhibitor Y27632 (StemCell Technology, cat. #72304)) and 20 ng/ml bFGF (Peprotech, cat. # AF-100-18B). Resuspended cells were then plated in 96-well ultra-low attachment plates (9000 cells per well) for formation of embryoid bodies (EBs). The hESC medium was renewed every other day until EBs reached 600-µm diameter, typically after 5 days. Then, EBs were transferred into 24-well ultra-low attachment plates and fed every other day with Neural Induction medium: 1% (v/v) N2 supplement, 1% (v/v) Gluta-MAX supplement, 1% (v/v) MEM-NEAA and 1 µg/mL heparin (Stemcell Technology, cat. #7980) in DMEM-F12 until neuroepithelium formation was apparent (usually after 4 days). EBs were then embedded into 20 µl Matrigel (Corning, cat. # 354230) droplets and transferred into a 60-mm tissue culture plate with 5 ml Cerebral Organoid Differentiation medium without vitamin A, prepared as following for 500 ml: 250 ml Neurobasal medium (Thermofisher, cat. # 21103049), 5 ml N2 supplement (Millipore-Sigma, cat. # M7145), 12.5 µl Insulin (Millipore-Sigma, cat. # I9278), 5 ml GlutaMAX, 5 ml MEM-NEAA, 5 ml penicillin-streptomycin (Thermofisher, cat. # 15140122) and 5 ml B27 without vitamin A (Thermofisher, cat. # 12587010) up to 500 ml with DMEM-F12, renewed every other day for 4 more days. Lastly, plates with the Matrigel-embedded organoids were transferred into an orbital shaker inside the incubator and fed with Cerebral Organoid Differentiation medium with vitamin A (same as previous but with B27 supplement with vitamin A (Thermofisher, cat. # 17504044)) for 5-6 days, renewing the medium every other day. Culture media contained 6–8 µM folic acid.

## Immunostaining of hiPSC-derived neural organoids

Neural organoids were fixed with 4% paraformaldehyde (PFA) at room temperature for 40 min, dehydrated and embedded in paraffin blocks. Sections 10 µm-thick were obtained, rehydrated, blocked with buffer containing 1% bovine serum albumin for 45 min at 23 °C and incubated with primary antibodies: FOLR1, 1:250 (Bioworld Technologies, cat. # BS386); SOX2, 1:200 (R&D Systems, cat. # AF2018); pan-cadherin, 1:300 (Abcam, cat. # ab22744); α-catenin, 1:250 (Invitrogen, cat. # 13-9700) in blocking buffer overnight at 4 °C, followed by incubation with fluorescently tagged Alexa Fluor secondary antibodies: anti-mouse 647 donkey, 1:300 (Invitrogen, cat. # A31571), anti-rabbit 488 donkey (Invitrogen, cat. # A21206) and anti-goat 596 donkey (Invitrogen, cat. # A11057) for 1 h at 23 °C. Imaging of 20–25 sections per fluorescently labeled neural organoid was done under a confocal microscope (Nikon A1) through a 1 µm-step z-stack scanning.

## FOLR1 knockdown in hiPSC-derived neural organoids

When hiPSC colonies reached 70–80% confluency, 2 µM standard control *vivo* morpholino, CCTCTTACCTCAGTTACAATTTATA (Control-MO, Control, Gene Tools, Inc.) or hFOLR1 translation blocking *vivo*-morpholino, GTTGTCATCCGCTGAGCCATGTC (FOLR1-MO, FOLR1 knockdown (KD), Gene Tools, Inc.) was added to the medium and incubated for 48 h. hiPSCs were then dissociated with Accutase solution (Sigma-Aldrich) and centrifuged for 3 min at 288 *g* to remove enzymatic solution. Cell pellet was resuspended in mTESR medium containing 5 µM ROCK inhibitor and 20 ng/ml bFGF, and then 9000 cells were seeded into Ultra low attachment surface 96-well plates. This was considered day 0, when FOLR1-MO-treated cultures were incubated either with freshly made 50 µM sodium pteroate or vehicle only.

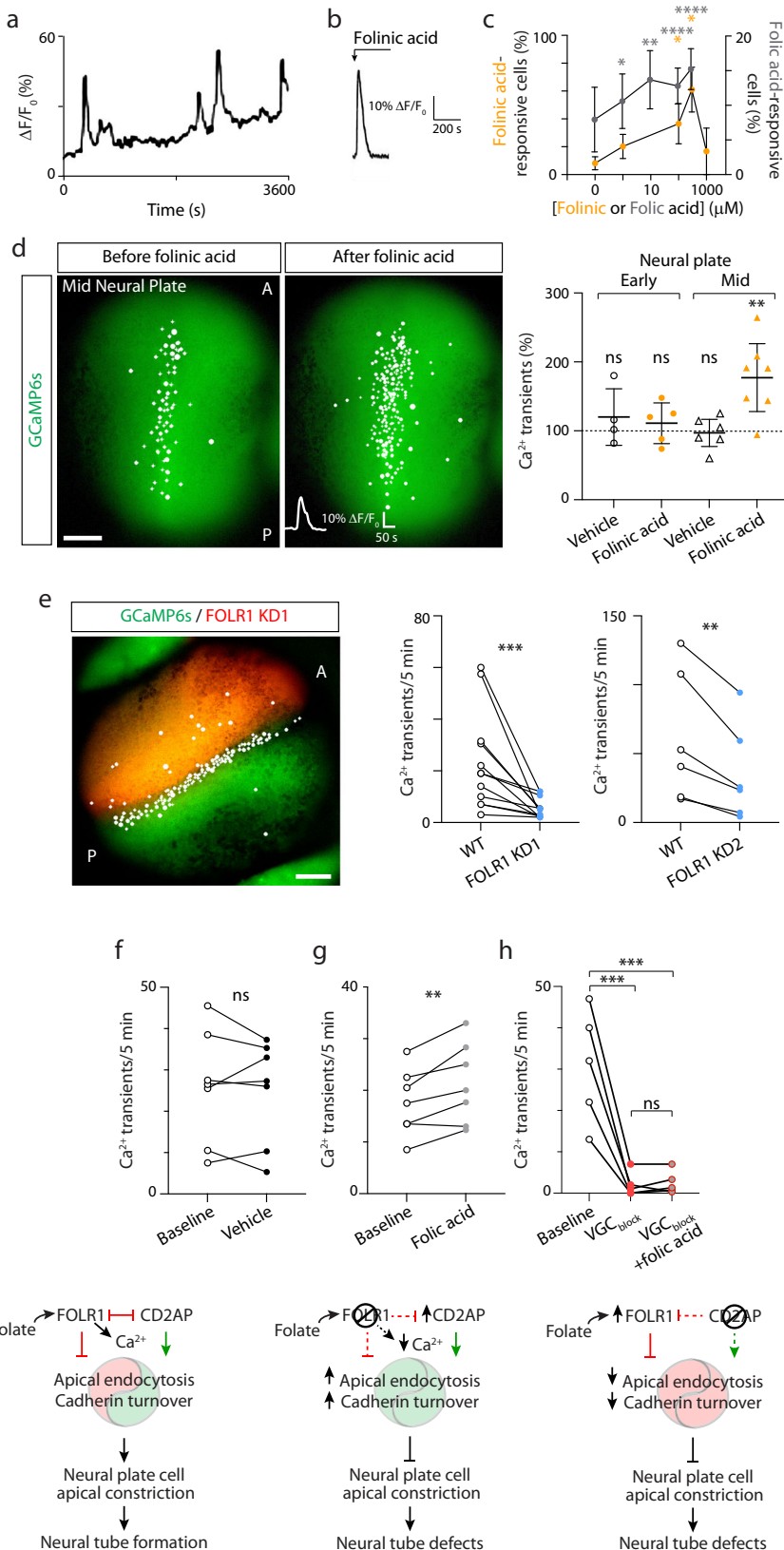

Pteroic acid (Santa Cruz, cat. # SC-250800) was dissolved in 0.05 N NaOH by vigorous shaking and solution was shielded from light. Treatments were renewed along with culture medium every other day.

To determine efficiency of FOLR1 KD, hiPSCs at 70% confluency were incubated with 2 μM Control-MO or FOLR1-MO for 48 h, collected and snap-frozen. Cells were thawed on ice, resuspended and lysed in

RIPA buffer, processed as described below for Western blot assays, and immunoblotted with anti-FOLR1 antibody, 1:500 in 5% BSA in TBST (Bioworld Technology, cat. # BS3861), HRP-conjugated secondary antibodies, 1:7500 (Jackson ImmunoResearch, cat. # 711-035-152) and ECL2 Western Blotting Substrate (ThermoFisher, cat. # 80196). PVDF membranes (cat. # IEVH00005, Millipore/Sigma) were stripped and

**Fig. 9 | Folate-FOLR1 contribute to Ca²⁺ dynamics in neural plate cells during neural plate folding. a–c** Neural plate from mid neural plate stage *Xenopus laevis* embryos was dissected and dissociated cells plated in vitro. After 2 h, cells were loaded with the Ca²⁺ sensor Fluo4-AM and time-lapse imaged. **a** Example of 1-h recording of neural plate cell Ca²⁺ activity. **b, c** Folinic acid (**b, c**), folic acid (**c**) or vehicle was added to neural plate cells in culture during time-lapse imaging and the Ca²⁺ response was recorded in the first minute post addition. **b** Example of acute transient elicited by 100 μM folinic acid. **c** Graph shows mean ± SEM folinic- or folic acid-responsive neural plate cells compared to total number of cells with spontaneous Ca²⁺ transients in 30 min recording, $N = 3$ independent experiments. Two-tailed one sample t and Wilcoxon test. **d** Two-cell stage embryos were bilaterally microinjected with GCaMP6s mRNA and grown until early and mid-neural plate stages when they were time-lapse imaged before and after addition of vehicle or 300 μM folinic acid. Image shows example of embryo with cells exhibiting Ca²⁺ transients indicated with circles. Graph shows individual and mean ± SD percent change in Ca²⁺ transient frequency before and after addition of vehicle or folinic

acid, $n = 4$, 5, 6 and 7 embryos for Early-Vehicle, Early-Folinic acid, Mid-Vehicle and Mid-Folinic acid groups, respectively. One-sample two-tailed t-test, compared to hypothetical value of 100%. **e–h** Two-cell stage embryos were bilaterally micro-injected with GCaMP6s mRNA (**e–h**) and unilaterally microinjected with 9.9 pmol FOLR1-MO1 (FOLR1 KD1/KD) or 1.6 pmol FOLR1-MO2 (FOLR1 KD2) per blastomere (**e**) and grown until mid-neural plate stages when they were time-lapse imaged. Image in (**e**) shows example of embryo with cells exhibiting Ca²⁺ transients indicated with circles in WT and FOLR1 KD1 halves. Graphs show individual Ca²⁺ transient frequency (transients/5 min) in WT and FOLR1 KD1 or KD2 halves (**e**, $n = 6$ embryos) and in WT embryos before and after addition of vehicle (**f**, $n = 7$ embryos), 50 μM folic acid (**g**, $n = 6$ embryos), Na⁺ and voltage-gated Ca²⁺ channel blockers (VGC$_{block}$: 0.02% tricaine+10 μM nitrendipine+25 μM TTA-2, **h**, $n = 5$ embryos), or a mix of folic acid and VGC$_{block}$ (**h**, $n = 5$ embryos). Two-tailed paired t-test (**e–g**) and 1-way ANOVA-Tukey multiple comparisons test (**h**). In (**c–h**), *$p < 0.05$, **$p < 0.01$, ***$p < 0.001$, ****$p < 0.0001$, ns: not significant. **i** Model of FOLR1 and CD2AP regulation of neural tube formation. Source data are provided as a Source Data file.

reprobed with anti-GAPDH antibody, 1:20,000 (Sicgen, cat. # AB0049-20) for loading control.

## FOLR1 knockout in hiPSC-derived neural organoids
Approximately 60% confluent hiPSCs were incubated with 1.2 μg targeted synthetic gRNA (AGUUGGGGGAGCACUCGUAG), 6.25 μg Cas9 and Lipofectamine CRISPRMAX Transfection Reagent (TrueCut Cas9 protein v2, Invtrogen) for 48 h. To assess efficiency of the KO-CRISPR approach, detection of cleavage was done by the GeneArt Genomic Cleavage Detection Kit (Invitrogen) according to the manufacturer's guidelines. Briefly, genomic DNA from transfected hiPSCs was extracted and the sequence of interest was amplified by PCR. After running the products in a gel, smaller size bands indicate indels introduced by Crispr-Cas9, while the largest band represents the WT DNA. Analysis of the gel bands was done by Image Lab software.

In addition, sequencing of genomic DNA was done to identify the % Indel and KO score. Analysis of the DNA sequences was done by the ICE Analysis online tool (Synthego).

## Analysis of immunostained neural organoids and neural tube-like structure defects
Large field of view images from 20–25 10 μm-thick sections of immunostained and DAPI-labeled neural organoids were used to record, in the section showing the largest extent of the complete organoid, the number of neural tube-like structures per organoid in control and experimental groups. This was based on gross appearance of tubularly-shaped structures and presence of central lumen.

**Nuclear displacement.** Neural organoid images of α-catenin immunostaining (for apical surface staining) and nuclear labeling by DAPI were used to measure the perpendicular distance from the apical surface of the neural tube surrounding the lumen to every nucleus of peri-lumen cells with the line tool in ImageJ software (NIH). At least 10 neural tube-like structures were measured for each group per experiment and at least 3 experiments were performed.

**Circularity measurement.** Neural organoid images of α-catenin immunostaining were used to measure the circularity of each peri-lumen cell of neural tube-like structures with the circularity tool in ImageJ software. The score is 1 for a perfect circle and close to 0 for elongated polygons.

**Cadherin intensity profile.** Neural organoid images of pan-cadherin immunostaining were used to measure the fluorescence intensity profile of cadherin immunostained neural tube-like structures by drawing a linear probe crossing the apical cell surface of the lumen with NIS Elements software (Nikon, Inc.). The peak values for cadherin fluorescence intensity, that correspond to the apical surface region

when apical constriction is apparent, was compared among control and experimental groups by calculating the percent change in cadherin fluorescence intensity when crossing the apical surface.

All these measurements were done in at least 10 neural tube-like structures for each group per experiment and at least 3 experiments were performed. Statistical analysis was done with Prism software (Graphpad) by using one-way ANOVA, Tukey's multiple comparisons test, $p < 0.05$ considered significant.

## *Xenopus laevis* animal handling and in vitro fertilization
Mature oocytes were collected in a dish from a previously hCG injected female frog and incubated with a small piece of minced testis. This is considered time 0 of fertilization. Fertilized oocytes were kept in 10% MMR saline, containing (in mM): 10 NaCl, 0.2 KCl, 0.1 MgSO₄, 0.5 Hepes, 5 EDTA and 0.2 CaCl₂. Dejellying of embryos was done by briefly swirling fertilized eggs in 2% cysteine solution, pH 8. Developmental stages were recorded according to Nieuwkoop and Faber[58]. Animals were handled according to the IACUC guidelines using humane procedures to prevent animal suffering.

## FOLR1 knockdown in *Xenopus laevis* embryos
Two-cell-stage embryos were unilaterally or bilaterally injected with 1.6–3.8 pmol splicing blocking morpholino (MO) complementary to *Xenopus laevis* folr1 exon-intron junction AAACCTTGGGCCCTG-GATCCAGAAGgtaattggaaggggggtgatggtgac, FOLR1-MO1 ATCACCCCCTTCCAATTACCTTCTG (FOLR1 KD1/KD) or with 9.9 pmol translation blocking MO (FOLR1 KD2) complementary to *Xenopus laevis* folr1 mRNA, used in previous study (11), FOLR1-MO2 GGCCCCCCCGTAA-CATGGTTACAAGC per blastomere. Controls were sibling embryos injected with standard control MO, Control-MO CCTCTTACCTCAGT-TACAATTTATA (Control). Morpholinos were injected along with dextran-Alexa-Fluor conjugates or with GFP or mCherry mRNA to assure permanency of MO reporter after TCA fixation. All sequences are written 5′ to 3′.

Rescue experiments were implemented by incubating embryos injected with 1.6 pmol FOLR1-MO1 per blastomere at 2-cell stage (4 h post fertilization (hpf) at 23 °C) with freshly made 50 μM sodium pteroate. FOLR1-MO-injected and control embryos were grown in the solution with pH matched to pteroate-incubated group. Embryos were grown in the dark at 18 °C overnight until stage 13, then transferred in freshly prepared solutions and grown up until early neural tube stages (stage 21, 22.5 hpf) at 23 °C.

Assessment of FOLR1 splicing blocking MO efficiency was performed by isolating neural plates from 9 stage 16 *Xenopus laevis* embryos in each group, resuspended in Trizol reagent (Invitrogen, cat. # 15596026) and stored at −80 °C. RNA was extracted with kit according to manufacturer's instructions (RNeasy Mini Kit, Qiagen, cat. # 74104), gDNA was eliminated (RapidOut DNA Removal Kit, Thermo

Scientific, cat. # 00859896) and cDNA was made (High-Capacity cDNA Reverse Transcription Kit, Applied Biosystems, cat. # 00890068) with standard protocols. Using this cDNA as template, PCR was performed for Control-MO and FOLR1-MO1 injected embryos with primers located in intron 2 and exon 3 that generates no PCR product from cDNA of mature, spliced *folr1* transcript and a 308 bp PCR product when the mRNA is not spliced. Forward primer: GCAAGACCCATTTTGTCCAG, Reverse primer: AGGCCACAGCCAATAGAAGA. All sequences are written from 5′ to 3′.

### Assessment of neural tube defect phenotype in *Xenopus laevis* embryos

Stereoscope images of embryos were taken when controls reached early neural tube stage (stage 20). Embryos were sorted into 2 groups: "Closed", those with apparently closed posterior neural tube and "NTDs", embryos with posterior neural tube open. To further characterize defects in the neural tissue, defective experimental embryos that survived past neural tube closure in control group and that did not show overt tissue degeneration were fixed with 4% PFA at 23 °C for 30 min and paraffin-embedded. Cellular, histological and morphological defects were assessed in 3 consecutive transverse sections of the embryo, stained with anti-SOX2, 1:200 (R&D Systems, cat. # AF2018), anti-α-tubulin, 1:500 (Abcam, cat. # ab15246) and DAPI (2 µg/ml), by assigning 1 point to the presence of each of the following defects: 1. Open neural tube (Supplementary Fig. 2b); 2. Absence of neural tube lumen (Supplementary Fig. 2c); 3. Non-folded neural plate (neural folds fall outside of the notochord lateral borders, Supplementary Fig. 2d); 4. Rounded neural cells (Supplementary Fig. 2d, e); 5. Neural tissue disruption (neural tissue breaks and loosely attached cells, Supplementary Fig. 2e). Neural tissue defect score was the median score of 3 transverse sections per embryo.

### Measurement of apical neural plate endocytosis in live intact *Xenopus laevis* embryos

Two-cell-stage embryos were bilaterally injected with 400 pg human EEA1 tagged to GFP (GFP-EEA1 wt was a gift from Silvia Corvera; Addgene plasmid # 42307; http://n2t.net/addgene:42307; RRID:Addgene_42307[59], subcloned into pcs2+), 200 pg membrane-mCherry mRNA (pCS-memb-mCherry was a gift from Sean Megason; Addgene plasmid # 53750; http://n2t.net/addgene:53750; RRID:Addgene_53750[60]) and unilaterally injected with 2.6–3.8 pmol of FOLR1-MO1 (FOLR1 KD) or 9.9 pmol of CD2AP-MO2 (CD2AP KD) or Control-MO (Control) along with Alexa 647-conjugated dextran. Medial neural plate of embryos at stage 14–14.5 were imaged using 60x objective and Z-stack confocal imaging (Sweptfield confocal, Nikon). Analysis was performed in time-lapse recordings with an acquisition rate of 1 frame/6 min, in total 3 frames. Number of endosomes and endosome/cell surface fraction were measured in FOLR1 KD, CD2AP KD, Control and contralateral wild type cells by creating a region of interest outlining cell boundaries and thresholding GFP intensity to at least 2 times over background with 0.4–4 µm size and 0.3–1 circularity limitations using NIS-Elements software (Nikon, Inc.).

### Immunohistochemistry of *Xenopus laevis* neural tissue sections

Stage 14–15 and 20–21 embryos were fixed at 23 °C with freshly made 2% trichloroacetic acid (TCA) or 4% PFA for 60 and 30 min, respectively, and processed for immunostaining as previously described with modifications and by using standard protocols of paraffin embedding and sectioning[16]. Incubations with primary and secondary antibodies were carried out overnight at 4 °C and for 2 h at 23 °C, respectively. Primary antibodies used were: anti-phospho-tyr-4,8,10 CD2AP, 1:500 (Rockland, cat. # 600-401-J96), anti-phospho-tyr-674 c-Cbl, 1:500 (MyBioSource, cat. # MBS820886), anti-C-cadherin, 1:100 (Developmental Studies Hybridoma Bank, cat. # 6B6), anti-EEA1, 1:1000 (Origene, cat. # AB0006-200), anti-ubiquitin, 1:500 (Stress Marq, cat. #

SPC-119B), anti-Rab7, 1:500 (Cell Signaling, cat. # 9367), anti-LAMP1, 1:500 (Abcam, cat. # ab24170), anti-GFP, 1:750 (Abcam, cat. # ab13970), anti-mCherry, 1:750 (Biorbyt, cat. # orb11618), anti-SOX2, 1:300 (Cat # AF2018, R&D Systems), anti-α-tubulin, 1:500 (Abcam, cat. # ab15246). Antigen retrieval was performed by boiling samples in 0.05% citraconic anhydride, pH 7.4[61]. Further processing starting with a 2% BSA blocking step was done using SNAP i.d. 2.0 System for immunohistochemistry (Millipore).

Quantitative assessment of C-cadherin-, EEA1-, ubiquitin-, Rab7-, LAMP1-containing vesicles was done by creating a region of interest outlining cell boundaries in transverse sections of the neural plate and thresholding immunolabeling fluorescence intensities followed by counting stained vesicles in 3 consecutive z-sections (1 µm step) with NIS Elements software (Nikon, Inc.). Samples were from early neural plate stage (stage14–14.5) embryos for C-cadherin/EEA1/ubiquitin and mid neural plate stage (stage 15) embryos for C-cadherin, Rab7 and LAMP1 immunostaining. At least 3 consecutive sections were analyzed per embryo from 5 embryos for each group.

### Western blot assays of *Xenopus laevis* embryo samples

To determine C-cadherin, FOLR1 and CD2AP protein levels, neural plates (2 per sample) were dissected from stage 15–17 *Xenopus laevis* wild type vehicle-/protein degradation inhibitor cocktail-treated embryos or Control/FOLR1 KD/CD2AP KD embryos in collagenase and snap-frozen in liquid nitrogen and stored frozen at −80°C overnight. Samples were thawed on ice and resuspended in RIPA buffer (1% (v/v) Triton X-100, 0.1% (w/v) SDS, 0.5% (w/v) Sodium deoxycholate, 150 mM NaCl, 25 mM Hepes pH 7.4, 2 mM EGTA, 2 mM EDTA), centrifuged at 16,100 $g$ for 10 min and pellet discarded. Protein loading buffer, 4X (200 mM Tris-HCl, pH 6.8, 8% (w/v) SDS, 40% (v/v) glycerol, 0.01% (w/v) bromophenol blue, 10% β-mercaptoethanol) was added to supernatant, followed by heating at 100 °C for 4 min. Samples were separated on 10% polyacrylamide gels and transferred to PVDF membranes (Millipore/Sigma, cat. # IEVH00005). Membranes were blocked with 0.5% non-fat milk TBST in SNAP i.d. 2.0 Protein Detection System (Millipore/Sigma) for 10 min and incubated with monoclonal anti-C-cadherin antibody, 1:100 (Developmental Studies Hybridoma Bank, cat. # 6B6) in 5% non-fat milk TBST overnight at 4 °C on rocking platform. Following 3x washes, membranes were incubated with HRP-conjugated secondary antibodies, 1:3000 (Jackson ImmunoResearch, cat. # 515-035-003,) in 0.5% milk TBST for 10 min in SNAP i.d. 2.0. for Western blot and visualized by ECL2 Western Blotting Substrate, (ThermoFisher, cat. # 80196). Signal was detected using ChemiDocMP Imaging System, Biorad. PVDF membranes were stripped in 0.2 M glycine HCL buffer, pH 2.5, 0.05% Tween for 20 min and re-probed with anti-CD2AP and anti-FOLR1 antibodies, 1:500 (custom made) in 5% milk TBST. Protein loading was detected using anti-GAPDH antibody, 1:5000 (Sicgen, cat. # AB0049-20) in 5% milk TBST, followed by incubation with HRP-conjugated secondary antibody, 1:7500 (cat. # 515-035-003, Jackson ImmunoResearch) and ECL Western Blotting Detection reagent (Millipore/Sigma, cat. # GERPN2106).

### Quantitative assessment of C-cadherin transcript expression by qPCR

Neural plates from 9 mid neural plate stage (stage 16) *Xenopus laevis* embryos in each group were isolated, resuspended in Trizol reagent (Invitrogen, cat. # 15596026) and stored at −80 °C. RNA was extracted with kit according to manufacturer's instructions (RNeasy Mini Kit, Qiagen, cat. # 74104), gDNA was eliminated (RapidOut DNA Removal Kit, Thermo Scientific, cat. # 00859896) and cDNA was made (High Capacity cDNA Reverse Transcription Kit, Applied Biosystems, cat. # 00890068) with standard protocols. Using this cDNA as template, qPCR was performed with SYBR Green Universal Master Mix (Applied Biosystems, cat. # 2107118) in Stratagene Mx3005 real-time PCR machine. RT-PCR program: 15 min 95 °C, 28 cycles of 45 s at 95 °C/30 s

at 55 °C/ 30 s at 72 °C, 1 min at 95 °C, 30 s at 55 °C and 30 s at 95 °C. C-cadherin forward primer: TGGTGACAGACGATGGTGTT, C-cadherin reverse primer: GCTGTCAAGTTCAGCCTTCC, ODC forward primer: GTCAATGATGGAGTGTATGGATC, ODC reverse primer: TCCATTCC GCTCTCCTGAGCAC. C-cadherin PCR product: 236 bp, ODC PCR product: 386 bp. All sequences are written from 5′ to 3′.

## Measurement of C-cadherin ubiquitination and cleavage

To determine the level of cleaved and ubiquitinated C-cadherin, stage 12–12.5 *Xenopus laevis* embryos were incubated with lysosome and proteasome inhibitor cocktail: 100 μM chloroquine (Selleckchem, cat. # 4157), 50 nM bafilomycin (Enzo Lifesciences, cat. # BML-CM110-0100) and 10 μM MG-132 (Selleckchem, cat. # 2619). At mid-neural plate stages (stage 16), neural plates (4 per sample) were dissected in collagenase, snap-frozen and stored at −80 °C for up to 2 weeks. Samples were thawed on ice and homogenized in 60 μl of lysis buffer containing 0.5% Sodium n-lauroylsarcosinate, 150 mM NaCl, 25 mM Tris pH 7.4, 2 mM EDTA, 2 mM EGTA, 100 μM PR-619 (Selleckchem, cat. # 7130) and protease inhibitors cocktail (Thermo Scientific, cat. # 784115,) by vortexing. After 10 min samples were diluted 4x with lysis buffer containing 1% Triton X-100, instead of 0.5% Sodium n-lauroylsarcosinate, centrifuged at 16,100 *g* for 10 min and pellet discarded. Supernatants were pre-cleared by incubating with 30 μl of protein G-agarose beads suspension (Roche, cat. # 11719416001) for 3 h at 4 °C on a mini-rotator. A quarter of the pre-cleared lysate was mixed with 4x reducing protein loading buffer (200 mM Tris-HCl, pH 6.8, 8% SDS, 40% glycerol, 0.01% bromophenol blue, 10% β-mercaptoethanol) and incubated at 100 °C for 4 min. Samples were separated on 10% poly-acrylamide gels and immunoblotted with anti-C-cadherin 1:100 (Developmental Studies Hybridoma Bank, cat. # 6B6), then reprobed with anti-GAPDH antibody (Sicgen, cat. # AB0049-20). The rest of the sample was incubated with protein G-agarose beads chemically cross-linked to anti-C-cadherin antibody (Developmental Studies Hybridoma Bank, cat. # 6B6) overnight at 4 °C on a mini-rotator. After 3x washing, beads were resuspended in 2X protein loading buffer containing 10 mM DTT and incubated for 10 min at 80 °C. Samples were separated on 10% polyacrylamide gels, immunoblotted with anti-ubiquitin antibody 1:500 in 5% BSA TBST (13-1600, Invitrogen) and reprobed with anti-C-cadherin antibody 1:100 in 5% non-fat milk TBST (Developmental Studies Hybridoma Bank, cat. # 6B6).

## LC-MS/MS analysis of FOLR1 immunoprecipitates

Fifteen mid-neural plate stage (stage 16) embryos were homogenized in lysis buffer containing 0.5% sodium n-lauroylsarcosinate, 150 mM NaCl, 25 mM Tris pH 7.4, 2 mM EDTA, 2 mM EGTA, and protease inhibitor cocktail on ice. Samples were diluted 4x with lysis buffer containing 1% Triton X-100, centrifuged at 16,100 *g* for 10 min and the pellet was discarded. Supernatants were pre-cleared by incubating with 30 μl of protein G-agarose beads (Roche, cat. # 11719416001) for 3 h at 4 °C on mini-rotator. FOLR1 was immunoprecipitated by overnight incubation of lysates with rabbit anti-*Xenopus laevis* FOLR1 polyclonal affinity purified antibody raised against the peptide KHQKVDPGPEDDLHC (custom made by GenScript), chemically cross-linked to protein G-agarose beads at 4 °C on a mini-rotator. Rabbit anti-GFP (TorryPines, cat. # TP401,) or normal rabbit IgG (Cell signaling, cat. # 2729) chemically cross-linked to protein G agarose beads were used as controls. Following overnight incubation, beads were washed 3x with 1% Triton X-100 lysis buffer and resuspended in non-reducing 2X protein loading buffer. A quarter of the sample was reduced by addition of 10 mM DTT and heated at 80 °C for 10 min, run on 10% polyacrylamide gel and immunoblotted for FOLR1. The rest of the non-reduced sample was run for 6 mm in resolving 10% polyacrylamide gels, stained with colloidal Coomassie, the protein region was excised, followed by in-gel trypsin digestion.

Digested peptides were analyzed by LC-MS/MS on a Thermo Scientific Q Exactive Plus Orbitrap Mass spectrometer in conjunction Proxeon Easy-nLC II HPLC (Thermo Scientific) and Proxeon nanospray source. The digested peptides were loaded in a 100 μm × 25 mm Magic C18 100 Å 5U reverse phase trap where they were desalted online before being separated using a 75 μm × 150 mm Magic C18 200 Å 3U reverse phase column. Peptides were eluted using a 100 min gradient with a flow rate of 300 nl/min. An MS survey scan was obtained for the m/z range 350–1600, MS/MS spectra were acquired using a top 15 method, where the top 15 ions in the MS spectra were subjected to HCD (High Energy Collisional Dissociation). An isolation mass window of 1.6 m/z was used for the precursor ion selection, and normalized collision energy of 27% was used for fragmentation. A 20 s duration was used for the dynamic exclusion.

Tandem mass spectra were extracted and charge state deconvoluted by Proteome Discoverer (Thermo Scientific). All MS/MS samples were analyzed using X! Tandem (The GPM, thegpm.org; version CYCLONE (2013.02.01.1)). X! Tandem was set up to search the uniprot-*Xenopus laevis* database (March 2017 version, 34096 entries) assuming the digestion enzyme trypsin. X! Tandem was searched with a fragment ion mass tolerance of 20 PPM and a parent ion tolerance of 20 PPM. Carbamidomethylation of cysteine was specified in X! Tandem as a fixed modification. Glu->pyro-Glu of the n-terminus, ammonia-loss of the n-terminus, gln->pyro-Glu of the n-terminus, deamidation of asparagine and glutamine, oxidation of methionine and tryptophan, dioxidation of methionine and tryptophan and acetyl of the n-terminus were specified in X! Tandem as variable modifications. Charge state deconvolution and deisotoping were not performed.

Criteria for protein identification. Scaffold (version Scaffold_4.8.4, Proteome Software Inc., Portland, OR) was used to validate MS/MS based peptide and protein identifications. Peptide identifications were accepted if they could be established at greater than 95.0% probability by the Scaffold Local FDR algorithm to achieve an FDR (Decoy) less than 1%. Protein identifications were accepted if they could be established at greater than 95.0% probability to achieve an FDR less than 5.0%. Protein probabilities were assigned by the Protein Prophet algorithm[62].

Proteomics experiment of FOLR1 co-IP was replicated 3 times; the proteins of interest were identified in all 3 LC-MS/MS runs.

The mass spectrometry proteomics data have been deposited to the ProteomeXchange Consortium via the PRIDE[63] partner repository with the dataset identifier PXD048476 and 10.6019/PXD048476. Project Name: FOLR1-interacting proteins in neurulating *Xenopus laevis* embryos. Project accession: PXD048476. Project DOI: 10.6019/PXD048476. Reviewer account details: Username: reviewer_pxd048476@ebi.ac.uk. Password: 0DFthm7n.

## Immunoprecipitation assays

Five stage 20 embryos bilaterally injected with 9.9 pmol CD2AP-MO2 (CD2AP KD) or Control-MO (Control, CD2AP immunoprecipitation (IP) experiments) and 15 stage 16 wild type embryos (FOLR1-CD2AP co-IP experiments) were collected and snap frozen. Samples were prepared as mentioned in previous section. One-tenth of the lysates was separated to assess for protein loading. IP/Western blot antibodies were rabbit anti-*Xenopus laevis* CD2AP polyclonal affinity purified antibody raised against the peptide CRPKSEVEPHSKTKT custom made by GenScript and anti-*Xenopus laevis* FOLR1 antibody (GenScript). After washing steps, protein G agarose beads with immunoprecipitates were resuspended in 2X protein loading buffer containing 10 mM DTT and incubated for 10 min at 80 °C. Samples were separated on 10% polyacrylamide gels and immunoblotted with anti-*Xenopus laevis* CD2AP antibody, 1:500 in 5% non-fat milk TBST followed by HRP-conjugated anti-rabbit secondary antibodies, 1:7,500 (GenScript, A01827-200) and ECL2 detection steps. Membranes were stripped and reprobed for FOLR1 antibody, 1:500. Protein loading was assessed in lysates with anti-β-tubulin, 1:300 (Developmental Studies Hybridoma Bank, cat. #

E7) in 5% non-fat milk TBST and FOLR1 antibody in 5% non-fat milk TBST (GenScript). Anti-GFP antibody or normal rabbit IgG were used as controls in FOLR1-CD2AP co-IP experiments.

## CD2AP knockdown in *Xenopus laevis* embryos

Two-cell-stage embryos were unilaterally or bilaterally injected with 2.6 pmol translation blocking CD2AP-morpholino 1 (MO1, CD2AP knockdown (KD) 1) TGTCACTCTCCGGCCTCTCGCTT, 7.4 – 9.9 pmol translation blocking CD2AP-MO2 (CD2AP KD2) CAATGTATTCCAC CATTCTGCTGCT complementary to *Xenopus laevis* CD2-associated protein (CD2AP) mRNA, or standard control-morpholino (Control-MO, Control, CCTCTTACCTCAGTTACAATTTATA), all from Gene Tools, Inc. Morpholinos were injected along with dextran-Alexa-Fluor conjugates or with GFP or mCherry mRNA to assure permanency of MO reporter in the injected side after TCA fixation.

Stereoscope images of embryos injected with 5.2 pmol CD2AP-MO1 or 14.8–19.8 pmol CD2AP-MO2 or Control-MO were taken when control embryos reached early neural tube stages (stage 20). Observed phenotypes were categorized in "Closed" or open ("NTDs") neural tube.

To assess CD2AP-MO efficiency, 5 *Xenopus laevis* embryos from control-MO- and CD2AP-MO2-injected groups were collected for immunoprecipitation assays to assess CD2AP protein level. Procedures of immunoprecipitation and Western blot assays were as described in previous section.

## Measurement of neural plate cell apical constriction

Apical constriction was assessed in stage 16.5 embryos unilaterally injected with 7.4–9.9 pmol of CD2AP-MO2 (CD2AP KD) and Control-MO (Control) in single blastomeres at the 2-cell stage. Measurements were done in 3 consecutive CD2AP KD and 3 paired Control medial neural plate cells in 5 consecutive sections from 4 embryos.

## *Xenopus laevis* neural plate cell cultures for in vitro $Ca^{2+}$ imaging

Mid-neural plate stage embryos (stage 15) were incubated with 1 mg/ml collagenase and neural plate was dissected. Neural plate cells were then dissociated by incubation with a $Ca^{2+}$/$Mg^{2+}$ free saline for 45 min. Cells were then plated in saline solution (in mM: 117 NaCl, 0.7 KCl, 1.3 MgSO4, 2 CaCl2, 4.6 Tris, pH 7.8), allowed to attach for 2 h and incubated for 45 min with 1 μM of the cell permeant $Ca^{2+}$ sensitive dye Fluo4-AM. Cultures were then time-lapse imaged in a Swept-field confocal microscope (Nikon) for up to 1 h at 0.2 Hz acquisition rate. Folic and folinic acid dose-dependent effectiveness in eliciting acute $Ca^{2+}$ transients in neural plate cells was assessed by addition of 1, 10, 100, 300 and 1000 μM folic or folinic acid and compared to addition of vehicle only (0 μM).

## In vivo $Ca^{2+}$ imaging

DNA encoding the $Ca^{2+}$ sensor GCaMP6s (pGP-CMV-GCaMP6s, a gift from Douglas Kim, (HHMI Janelia Research Campus, Ashburn, Virginia); plasmid #40753, Addgene[64]) was subcloned into the pCS2 vector using BglII and NotI restriction sites. The BglII restriction site was included in pCS2 with the following primers: forward, 5'-TCACTA AAGGGAACAAAAGATCTGGTACCGGGCCCAA-3'; reverse, 5'-TTGGGC CCGGTACCCAGATCTTTTGTTCCCTTTAGTGA-3'. mRNA was transcribed using mMessage mMachine kit (Ambion). GCaMP6s mRNA was injected in 1 to 4-cell stage embryos (1 ng mRNA/embryo). Neural plate stage embryos (stage 13–17; 14–19 hpf) were imaged under a macroscope (mode AZ100, Nikon) at an acquisition rate of 0.1 Hz for 20 min. Detection of $Ca^{2+}$ transients was thresholded by a peak change in fluorescence of at least 2 times the noise, as in previous studies[43,65–67]. The number of $Ca^{2+}$ transients at different developmental stages in the presence or absence of 300 μM folinic acid or vehicle only, in unilaterally injected with FOLR1-MO, or in the presence of ion channel blockers and folic acid in wild type embryos were measured, and significance was assessed by paired *t* test or one-way ANOVA followed by

Tukey's multiple comparisons test. To assess the $Ca^{2+}$ source of folic acid-induced $Ca^{2+}$ transients GCaMP6s-expressing embryos were live imaged for a maximum of 30 min under a confocal microscope (Nikon A1): first 5 min to record baseline activity, followed by addition of a $Na^+$ and $Ca^{2+}$ channel blocker cocktail ($VGC_{block}$): 0.02% tricaine, $Na^+$ channel blocker + 10 μM nitrendipine, L-type voltage-gated $Ca^{2+}$ channel blocker + 25 μM TTA-2, T-type voltage-gated $Ca^{2+}$ channel blocker, for 10 min and, of $VGC_{block}$ + 50 μM folic acid for 15 min.

## Statistics and reproducibility

Rigorous research design and analysis was implemented by running all the controls necessary alongside experimental samples. Analysis of data was performed blindly to the analyzer with the exception of analysis of Western blot assays that results were obtained directly from the developer by the experimenter running the gels who knew the order in which samples were loaded. Number of samples for each experiment was determined by pilot experiments and power analysis. Experiments were replicated at least 3 times. No data were excluded from the analysis. Statistical analysis of the data was done with Prism software (Graphpad, Inc.). Normality test was performed in each set of data and then parametric (normally distributed) or non-parametric statistical analysis was chosen. Paired tests were implemented in unilaterally manipulated embryos, when compared control and microinjected halves of neural tissue or when $Ca^{2+}$ activity was compared before and after addition of an agent in the same sample. Number of samples analyzed per group were more than 5. Groups were considered statistically different when $p < 0.05$. Exact $p$ values are included in the Source Data File for data set. Each statistical test used is indicated in each figure legend for every dataset.

## Reporting summary

Further information on research design is available in the Nature Portfolio Reporting Summary linked to this article.

## Data availability

All data supporting the findings of this study are available within the paper and its Supplementary Information. The mass spectrometry proteomics data have been deposited to the ProteomeXchange Consortium via the PRIDE[63] partner repository with the dataset identifier PXD048476 and https://doi.org/10.6019/PXD048476. Source data are provided as a Source Data file with this paper. Source data are provided with this paper.

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

## Acknowledgements

We thank Andrew Hamilton for comments on the manuscript. This work was supported by: National Science Foundation grant 1754340 (L.N.B.), National Institutes of Health grants R01NS105886 and R01NS113859 (L.N.B.), Shriners Hospital for Children grant 85111 (L.N.B.) and grant 84306 (O.A.B.).

## Author contributions

Conceptualization: O.A.B., L.N.B. Methodology: O.A.B., A.A.P., O.V., J.S. Investigation: O.A.B., A.A.P., O.V. Funding acquisition: O.A.B., L.N.B., P.S.K. Project administration: L.N.B. Supervision: L.N.B., P.S.K. Writing – original draft: O.A.B., A.A.P., L.N.B. Writing – review & editing: O.A.B., A.A.P., L.N.B.

## Competing interests

The authors declare no competing interests.
