## [Peer Review File · Nature Communications]

Noncanonical function of folate through folate receptor 1 during neural tube formationREVIEWER COMMENTS

Reviewer #1 (Remarks to the Author):

Although the occurrence of NTDs in humans has long been epidemiologically thought to be suppressed by daily folic acid (FA) intake, the molecular mechanism was unknown, while the chemical properties of the compound suggested that metabolic function of FAD might be involved. Most intriguingly and importantly, this study revealed a possible mechanism in which a folate receptor and its counteracting protein play critical roles in the regulation of cadherin turnover and proposed a novel non-metabolic function of folate. This unique work full of compelling data would have the potential to enrich and advance the field of NTD research.

Major points

1. It is described that pteroylserine, a folate precursor that binds to FOLR1 was used to rescue the neural tube defect phenotype. It is suggested that the authors may have expected to eliminate the contribution of metabolic pathways. However, one might expect that folate, an endogenous ligand for FOLR1 would be sufficient to rescue the defective phenotype unless the metabolic function that folate induces does not inhibit the rescue. The authors need to show the results of the rescue experiment using the authentic ligand folate.
2. As the authors' proposal for the mechanism of cell behavior by folate is novel and of high impact, it is important to address how generally the mechanism is adopted for other organogenesis. The authors should provide more information about the generality of the mechanism given the broad expression of FOLR1 and CD2AP in tissues. Namely, is this model specific to the neural tube or is it generalized to other tissues?
3. Line 222, the finding of the acute Ca^{2+} transients induced by folates is impressive. However, can it be interpreted as a result of the effects of folates on cadherin endocytosis? The onset of Ca^{2+} transients seems to be too fast for the time required to regulate cadherin endocytosis. Although it is true that attenuation of cadherin function eventually alters Ca^{2+} dynamics, the authors need to explain this or discuss the acute Ca^{2+} transient as an event independent from the endocytosis.

Minor points

1. Line 91, it is better to be: apico-basally elongate
2. The use of chemical names, should be clarified and unified if it is possible. For example, line 222: folates, 227: folinates.

Reviewer #2 (Remarks to the Author):

Title: Non-canonical function of folate/folate receptor 1 during neural tube formation

Authors: Balashova, O.A. et al.,

Summary: This manuscript describes a series of experiments in human cell derived neural organoids and in *Xenopus Laevis* models that the folate receptor (Folr1) is essential for the formation of the neural tube and that it is possible to rescue abnormal neurulation by the administration of pteroylserine which binds to FOLR1. This protein transporter opposes the function of CD2-associated protein (CD2AP), whose expression is required for apical endocytosis and turnover of the cell adhesion C-cadherin in the early neurulation stage embryo. The authors suggest that folate/FOLR1 regulate neural

plate folding and its disruption is the mechanism underlying abnormal neural tube development. The approach is novel, as the investigators used a folate precursor (pteroate) that binds to folate receptor (FOLR1) but does not contribute to one carbon metabolism. Overall, the paper is well-written and organized; however, but needs better clarification on a few major points.

Major concerns:

1. The manuscript presents evidence from neural organoids and *Xenopus* embryos that FOLR1 regulates apical endocytosis and turnover of cadherin during neural plate folding. It also reports that pteroate binds to FOLR1 and triggers calcium transients that are necessary for neural plate cell apical constriction. Unfortunately, it does not report how calcium would enter the cell or embryo. Is this being controlled by a voltage-gated calcium channel? The observed mechanism for this change in calcium should be identified. If this is unknown, then such experiments must be performed as they are critical to properly evaluate the mechanism suggested in the text.
2. GCaMP6s measures intracellular calcium, but is the calcium being accumulated from outside of cells and/or being passaged-compartmentalized differently inside cells? Were there "waves" moving calcium towards the midline of the organoids? Again, this must be clarified in the revised text.
3. The authors should explain why pteroate rescued some aspects of neural development (Figs. 1b, 1c, 1d), but not other changes (Fig. 1e). Are some aspects one carbon dependent and others are not, or is it all related to the amount of residual FOLR1? Did the authors look at FOLR2? It is mentioned in the Introduction of the manuscript, but does it not have the same function to rescue, only with a lower affinity, or is it not expressed at this time? This is important to understanding the underlying mechanism at the core of the manuscript.
4. Figure 7 includes folinate, folate, and pteroate. Which folate is being referenced by folate? Is that folic acid? There is no other reference to the use of folic acid outside of cell culture, so folate is unclear in this experiment/figure. Why is there no concentration of folate in figure 7C at 1mM? Why is there a 1mM in that figure if concentrations tested were 0-500uM? This must be clarified in any revision of this manuscript.

Minor concerns:

1. Introduction: The authors question whether folate supplementation prevents NTDs via its role in one carbon metabolism or through other mechanisms and suggest that this has not been "fully investigated". It is insulting to this reviewer that the authors choose to neglect a robust literature of well over 3400 publications. I hope that these investigators revise this sentence in any potential revision of this submitted manuscript to recognize the fact that they are not opening a door that has not previously been opened.
2. Introduction: The authors erroneously state that the proton-coupled folate transporter (PCFT) is restricted to the intestinal epithelial. If they had read any of the aforementioned literature, they would know that PCFT is expressed in the choroid plexus (Alam C, et al., Upregulation of reduced folate carrier by vitamin D enhances brain folate uptake in mice lacking folate receptor alpha. *Proc Natl Acad Sci U S A.* 2019 Aug 27;116(35):17531-17540. doi: 10.1073/pnas.1907077116.
3. Results: Just because of the location of FOLR1 in neural cells in human organoids and in the frog neural plate, the authors claim that there is conservation of apical targeting in cells forming the neural tube. This is quite the leap of faith that deserves more critical analysis by the authors.

Dear Reviewers,

Please find below our point-by-point responses (in purple) to your critiques (in italics) of the manuscript entitled "**Non-canonical function of folate/folate receptor 1 during neural tube formation**". We thank the reviewers for their careful evaluation of our manuscripts and their thoughtful critiques and suggestions.

We believe that the additional experiments we performed, new data included in this manuscript, along with the extensive revision we have done to the text, have addressed satisfactorily the reviewers' concerns and resulted in a much-improved study.

Thank you for your consideration of our manuscript.

Laura Borodinsky

Professor

UC Davis

Reviewer #1

Although the occurrence of NTDs in humans has long been epidemiologically thought to be suppressed by daily folic acid (FA) intake, the molecular mechanism was unknown, while the chemical properties of the compound suggested that metabolic function of FAD might be involved. Most intriguingly and importantly, this study revealed a possible mechanism in which a folate receptor and its counteracting protein play critical roles in the regulation of cadherin turnover and proposed a novel non-metabolic function of folate. This unique work full of compelling data would have the potential to enrich and advance the field of NTD research.

We thank the reviewer for their appreciation of our manuscript and the impact it may have in the field.

Major points

- 1. It is described that pteroate, a folate precursor that binds to FOLR1 was used to rescue the neural tube defect phenotype. It is suggested that the authors may have expected to eliminate the contribution of metabolic pathways. However, one might expect that folate, an endogenous ligand for FOLR1 would be sufficient to rescue the defective phenotype unless the metabolic function that folate induces does not inhibit the rescue. The authors need to show the results of the rescue experiment using the authentic ligand folate.*

We have reported in a previous study (Balashova et al., Development 2017) the rescue of FOLR1 knockdown-induced neural tube defect phenotype by incubating embryos with folinic acid – metabolically active reduced form of folic acid. We refer to these findings in this manuscript when we compare them to the rescue by pteroate, presented in this study. We have revised the text to make this point clearer (page 4, last paragraph).

We have also included a reference to a published study demonstrating the absolute requirement of the pterin group for anchoring folate in the binding pocket of the receptor (Chen et al, Nature 2013, reference 21). On the other hand, the lack of a glutamate group disables pteronic acid as a substrate for metabolic

enzymes. Hence, in this study we strategically chose pterotic acid to test our hypothesis that it is not the restoring of folate metabolism what rescues the neural tube defect induced by FOLR1 knockdown when samples are incubated with a FOLR1 ligand.

2. *As the authors' proposal for the mechanism of cell behavior by folate is novel and of high impact, it is important to address how generally the mechanism is adopted for other organogenesis. The authors should provide more information about the generality of the mechanism given the broad expression of FOLR1 and CD2AP in tissues. Namely, is this model specific to the neural tube or is it generalized to other tissues?*

This is an interesting point. The role of CD2AP in regulating cell-cell adhesion in several tissues is well recognized, most notably in kidneys. However, in kidneys CD2AP and FOLR1 are localized to different regions. CD2AP is found in developing podocytes and glomerular basal membrane as well as in the epithelium of collecting ducts. FOLR1 is highly expressed in microvilli of proximal tubules. In contrast, this is the first study that shows apical colocalization of FOLR1 and CD2AP in neural plate cells and their interaction and how these two proteins counteract each other in regulating cell-cell adhesion during neural tube formation. Interestingly, we discovered that de novo synthesis of FOLR1 and its localization to the apical membrane of neural plate cells is necessary for neural tube formation, suggesting that localized expression of FOLR1 and CD2AP may be timely regulated to enable specific cell behavior and tissue shaping.

To be noted, while expression of the main folate transporter, the reduced folate carrier, is ubiquitous, FOLR1 expression is not ubiquitous. Instead, during neural tube formation, is enriched in neural folds as shown in mouse embryos and is enriched apically in neural plate cells in *Xenopus laevis* embryos. It is also expressed in the choroid plexus where it serves as important folate transporter for the neural tissue.

We revised the text to better position our findings in a broader context of organogenesis and apical constriction in other tissues (Discussion, page 8, 2nd paragraph)

3. *Line 222, the finding of the acute Ca²⁺ transients induced by folates is impressive. However, can it be interpreted as a result of the effects of folates on cadherin endocytosis? The onset of Ca²⁺ transients seems to be too fast for the time required to regulate cadherin endocytosis. Although it is true that attenuation of cadherin function eventually alters Ca²⁺ dynamics, the authors need to explain this or discuss the acute Ca²⁺ transient as an event independent from the endocytosis.*

We appreciate the reviewer's thoughtful interpretation of the potential link between cadherin endocytosis and folate contribution to Ca²⁺ signaling in the neural plate. We agree that the understanding of the specific sequence and nature of the connection between these two phenomena requires further investigation, which lies outside of the scope of this study. Nevertheless, we do believe that previous studies have demonstrated an association between Ca²⁺ activity and apical constriction through coordinated pulses of actomyosin contraction. In turn, these events of actomyosin contraction are coordinated with apical membrane endocytosis and adherens junction remodeling.

Often, Ca²⁺ signaling operates through spatiotemporal integration of the Ca²⁺ transient events occurring in a cell or tissue, which then results in the recruitment of a specific signaling cascade that leads to a certain cellular outcome. Given that our model supports FOLR1 inhibiting cadherin endocytosis by opposing CD2AP action through triggering its degradation, a plausible mechanism for FOLR1-induced downregulation of CD2AP may consist in enhancing Ca²⁺ activity in the neural plate that results in the recruitment of a Ca²⁺-sensitive protease, like calpain. Interestingly, calpain activity is necessary for neural tube formation (Christodoulou and Skourides, 2015). We revised the discussion to include further explanation on how FOLR1-induced Ca²⁺ signaling may regulate cadherin endocytosis during neural plate folding. We also revised the manuscript to indicate that folate/FOLR1-Ca²⁺ signaling and folate/FOLR1-regulation of cadherin endocytosis may be indirectly related events (Discussion, page 9, first paragraph).

Minor points

1. *Line 91, it is better to be: apico-basally elongate*

We revised the sentence according to the reviewer's suggestion.

2. *The use of chemical names, should be clarified and unified if it is possible. For example, line 222: folates, 227: folinates.*

We have referred to folate when talking generically about the family of folate derivatives or when referring specifically to the base of folic acid, and to folinate when referring specifically to the base of folinic acid. Nonetheless, we agree with the reviewer that it is confusing. We revised the manuscript to refer to folates as generic for folate derivatives and to folic acid or folinic acid when using those specific forms.

Reviewer #2

The approach is novel, as the investigators used a folate precursor (pteroate) that binds to folate receptor (FOLR1) but does not contribute to one carbon metabolism. Overall, the paper is well-written and organized; however, but needs better clarification on a few major points.

We thank the reviewer for their appreciation of our manuscript.

Major concerns:

1. *The manuscript presents evidence from neural organoids and Xenopus embryos that FOLR1 regulates apical endocytosis and turnover of cadherin during neural plate folding. It also reports that pteroate binds to FOLR1 and triggers calcium transients that are necessary for neural plate cell apical constriction. Unfortunately, it does not report how calcium would enter the cell or embryo. Is this being controlled by a voltage-gated calcium channel? The observed mechanism for this change in calcium should be identified. If this is unknown, then such experiments must be performed as they are critical to properly evaluate the mechanism suggested in the text.*

We performed additional experiments to address the reviewer's inquiry about the source of folate-induced Ca^{2+} activity and find that these transients depend on Ca^{2+} influx. In particular, we find that they are dependent on Na^+ channels and T- and L-type voltage-gated Ca^{2+} channels. We included these data in the revised Figure 7f-h.

Several studies have shown multiple mechanisms contributing to Ca^{2+} dynamics in the neural plate. For instance, our lab has shown that glutamate signaling through NMDA receptors contributes, at least partially, to Ca^{2+} activity in the neural plate (Sequerra, Goyal, et al., 2018). Other labs have also demonstrated that L-type voltage-gated Ca^{2+} channels participate in Ca^{2+} dynamics in the neural plate (Suzuki et al., 2017). These are all examples of plasma membrane Ca^{2+} -permeable channels indicating that Ca^{2+} transients in the neural plate are dependent on Ca^{2+} influx.

2. *GCaMP6s measures intracellular calcium, but is the calcium being accumulated from outside of cells and/or being passaged-compartmentalized differently inside cells?*

The new data we collected in response to the reviewer's inquiry indicate that Ca^{2+} transients are dependent on Ca^{2+} influx. However, Ca^{2+} influx may, in turn, trigger release from stores, contributing to the overall amplitude of Ca^{2+} transient, as it is often the case in developing neural cells. We have revised the text to clarify this point as suggested by the reviewer and included the new data in Figure 7f-h.

Were there "waves" moving calcium towards the midline of the organoids? Again, this must be clarified in the revised text.

Most of the observed Ca^{2+} transients occur in a cell autonomous and asynchronous manner (Christodoulou and Skourides, 2015). Nevertheless, the frequency of these transients increases with the progression of neural plate folding (Sequerria, Goyal, et al., 2018). We revised the text to clarify this point (Discussion, page 7, 2nd paragraph).

3. *The authors should explain why pteroate rescued some aspects of neural development (Figs. 1b, 1c, 1d), but not other changes (Fig. 1e). Are some aspects one carbon dependent and others are not, or is it all related to the amount of residual FOLR1?*

The difference between Fig. 1b-d and Fig. 1e is that for 1b-d we are using a FOLR1-knockdown approach, while for 1e is a FOLR1-knockout approach of hiPSC-derived neural organoids, which suggests, as the reviewer proposes that the residual FOLR1 is what enables rescue in the knockdown model and not in the knockout. Indeed, we chose this research design to demonstrate that pteroate rescue is dependent on a threshold level of FOLR1 protein expression remaining per neural cell, that is reached in FOLR1-KD but not in FOLR1-KO samples. Thus, pteroate rescues the defective neural tube and depleted cadherin phenotype in KD (Fig. 1b-d and Fig. 2a) but not in KO (Fig. 1e and Fig. 2b) samples.

Did the authors look at FOLR2? It is mentioned in the Introduction of the manuscript, but does it not have the same function to rescue, only with a lower affinity, or is it not expressed at this time? This is important to understanding the underlying mechanism at the core of the manuscript.

We thank the reviewer for their inquiry on FOLR2. However, the lack of rescue of the defective neural organoid phenotype by pteroate in FOLR1-KO unlike in FOLR1-KD samples suggests that there is no redundancy in FOLR1 and FOLR2 function in this model system, and that FOLR1 expression is necessary for pteroate rescue.

This is similar to the lack of rescue of the NTD phenotype in folbp1-KO (folbp1^{-/-}) mice with folic acid supplementation and absence of phenotype in folbp2-KO mice, while in folbp1+/- mice the lethality phenotype is rescued by supplementing with folinic acid (Piedrahita et al., 1999). This suggests that FOLR2 is either not expressed at those developmental stages or not expressed in the required spatiotemporal pattern for neural tube formation to substitute for FOLR1 expression.

Nonetheless, we performed additional experiments to determine folr2 expression in hiPSC-derived neural organoids and find that folr2 is not expressed during the period of evaluation in this study, which is during the formation of neural tube-like structures (See Figure below).

FOLR2 is not expressed in hiPSC-derived neural organoid cultures.

Shown is a PCR assay from cDNA collected from WT hiPSCs using specific primers for FOLR2 and GAPDH and from a synthesized oligonucleotide corresponding to the FOLR2 sequence amplified by the primers, as a positive control (expected size 101 bp). Three experiments showed similar results.

4. *Figure 7 includes folinate, folate, and pteroate. Which folate is being referenced by folate? Is that folic acid? There is no other reference to the use of folic acid outside of cell culture, so folate is unclear in this experiment/figure.*

We apologize for the confusion in the naming of folates. Yes, in the original Figure 7 we referred to folate as the base of folic acid. We revised the manuscript to refer to folates as generic for folate/folate derivatives and to folic acid or folinic acid when using those specific forms of folate.

Why is there no concentration of folate in figure 7C at 1mM? Why is there a 1mM in that figure if concentrations tested were 0-500uM? This must be clarified in any revision of this manuscript.

With regard to Figure 7c, we find that the maximum response in Ca²⁺ activity elicited in neural plate cells is 100 uM, similar to that elicited by 300 uM folic acid, thus, probing the samples with 1 mM was not necessary. Instead, with folinic acid, the effect of 300 uM was larger than with 100 uM, thus we tried 1 mM folinic acid to identify the maximal concentration, which it turns out to be 300 uM.

Because the x axis in the graph in Figure 7C is shared by the two curves, for folic and folinic acid, we need to have a 0 to 1 mM scale, despite folic acid was tested at a maximum of 300 uM.

Minor Concerns

1. *Introduction: The authors question whether folate supplementation prevents NTDs via its role in one carbon metabolism or through other mechanisms and suggest that this has not been “fully investigated”. It is insulting to this reviewer that the authors choose to neglect a robust literature of well over 3400 publications. I hope that these investigators revise this sentence in any potential revision of this submitted manuscript to recognize the fact that they are not opening a door that has not previously been opened.*

We apologize for misconstruing the referred sentence. It was not our intention to dismiss the body of work and literature dedicated to investigating the mechanisms of action of folate supplementation in preventing NTDs. We have revised this sentence and included more references to represent accurately the past and current efforts in the field and the current status of understanding on this topic (Introduction, first paragraph).

2. *Introduction: The authors erroneously state that the proton-coupled folate transporter (PCFT) is restricted to the intestinal epithelial. If they had read any of the aforementioned literature, they would know that PCFT is expressed in the choroid plexus (Alam C, et al., Upregulation of reduced folate carrier by vitamin D enhances brain folate uptake in mice lacking folate receptor alpha. Proc Natl Acad Sci U S A. 2019 Aug 27;116(35):17531-17540. doi: 10.1073/pnas.1907077116.*

We apologize for this oversight. We corrected the text and added the reference suggested by the reviewer as well as others that effectively review the theme of folate transporters more broadly.

3. *Results: Just because of the location of FOLR1 in neural cells in human organoids and in the frog neural plate, the authors claim that there is conservation of apical targeting in cells forming the neural tube. This is quite the leap of faith that deserves more critical analysis by the authors.*

We revised the text to avoid over concluding (Results, 1st paragraph). Nonetheless, GPI-anchored proteins are frequently targeted to apicobasal membranes in a polarized manner to exert specialized functions. Hence, the fact that we discovered similar localization in neural cells in two different model systems is suggestive of a conserved subcellular localization for FOLR1 during neural tube formation.

REVIEWERS' COMMENTS

Reviewer #1 (Remarks to the Author):

The authors have faithfully addressed the points which this reviewer raised and improved the manuscript.

Reviewer #2 (Remarks to the Author):

This reviewer is satisfied with the revised manuscript that included new experiments to resolve issues/concerns about calcium that were raised in the initial review. The comments and explanations demonstrate a sufficient level of understanding of the desired changes that are, in fact, reflected in the revised manuscript.